# Robust Sparsification via Sensitivity

**Chansophea Wathanak In** [1]   **Yi Li** [1 2]   **David P. Woodruff** [3]   **Xuan Wu** [1]

## Abstract

Robustness to outliers is important in machine learning. Many classical problems, including subspace embedding, clustering, and low-rank approximation, lack scalable, outlier-resilient algorithms. This paper considers machine learning problems of the form $\min_{x \in \mathbb{R}^d} F(x)$, where $F(x) = \sum_{i=1}^{n} F_i(x)$, and their robust counterparts $\min_{x \in \mathbb{R}^d} F^{(m)}(x)$, where $F^{(m)}(x)$ denotes the sum of all but the $m$ largest $F_i(x)$ values. We develop a general framework for constructing $\varepsilon$-coresets for such robust problems, where an $\varepsilon$-coreset is a weighted subset of functions $\{F_1(x), \ldots, F_n(x)\}$ that provides a $(1+\varepsilon)$-approximation to $F(x)$. Specifically, if the original problem $F$ has total sensitivity $T$ and admits a vanilla $\varepsilon$-coreset of size $S$, our algorithm constructs an $\varepsilon$-coreset of size $\tilde{O}(\frac{mT}{\varepsilon}) + S$ for the robust objective $F^{(m)}$. This coreset size can be shown to be near-tight for $\ell_2$ subspace embeddings. Our coreset algorithm has scalable running time and, by employing a sensitivity flattening argument, leads to new or improved algorithms for robust optimization problems, including regression and PCA. Finally, empirical evaluations demonstrate that our coresets outperform uniform sampling on real-world data sets.

## 1. Introduction

Outliers, which can occur frequently in real-world data, pose significant challenges to many machine learning problems. For instance, in the classic regression problem, a small number of contaminated data points can drastically skew the solution. This issue has driven the development of robust regression methods since the 1960s (see, e.g., (Andersen, 2008) for a comprehensive survey), and it remains an active area of research today. In this paper, we present a framework to address the issue of outliers for a large family of optimization problems, including regression, by reducing their scale through the construction of coresets.

A natural and popular approach for handling outliers is to consider robust versions of the original optimization formulations. Specifically, consider an optimization problem of the form $\min_{x \in \mathbb{R}^d} F(x)$, where $F(x) = \sum_{i=1}^{n} F_i(x)$. The robust version of $F(x)$, denoted by $F^{(m)}(x)$, aggregates all but the largest $m$ values over $F_i(x)$ ($i = 1, \ldots, n$). Formally, for each $x \in \mathbb{R}^d$, the values $F_1(x), \ldots, F_n(x)$ are sorted in increasing order as $F_{i_1}(x) \geq F_{i_2}(x) \geq \cdots \geq F_{i_n}(x)$, and $F^{(m)}(x)$ is then defined as $F^{(m)}(x) = \sum_{j=1}^{n-m} F_{i_j}(x)$. This robust formulation coincides with the least trimmed regression (Rousseeuw, 1985) and was first considered in algorithmic machine learning by (Charikar et al., 2001) in the context of facility location. More recently, it has been studied in several important machine learning problems, including PCA (Simonov et al., 2019) and clustering (Chen, 2008; Krishnaswamy et al., 2018). However, all those algorithms have high-order polynomial running times, making them neither scalable nor even practical.

In the case of no outliers (which will be referred to as the *vanilla* case), a popular scalable solution is to sparsify the input by constructing a coreset. A (strong) $\varepsilon$-coreset is a weighted subset of the dataset such that the loss of every candidate solution on this subset approximates the original loss within a relative error of $\varepsilon$. In the past 20 years, coresets have been extensively studied for a variety of machine learning problems (Har-Peled & Mazumdar, 2004; Feldman & Langberg, 2011; Munteanu et al., 2018; Chhaya et al., 2020; Jubran et al., 2021; Huang et al., 2021; Braverman et al., 2022).

In the robust case, early coresets have either an exponential size (Feldman & Schulman, 2012) or a weaker bi-criteria approximation guarantee (Feldman & Langberg, 2011; Huang et al., 2018). Recently, in the context of clustering with outliers, Huang et al. (Huang et al., 2023a) construct the first strong coreset with a similar size to the vanilla case. Inspired by their work, more strong robust coresets for clustering have been developed (Huang et al., 2025; Jiang &

The authors are listed in alphabetical order. [1]School of Physical and Mathematical Sciences, Nanyang Technological University, Singapore [2]College of Computing and Data Sciences, Nanyang Technological University, Singapore [3]Department of Computer Science, Carnegie Mellon University, USA. Correspondence to: Yi Li <yili@ntu.edu.sg>.

*Proceedings of the 42nd International Conference on Machine Learning*, Vancouver, Canada. PMLR 267, 2025. Copyright 2025 by the author(s).

Lou, 2025). Beyond clustering, Wang et al. (Wang et al., 2021) constructed local robust coresets for continuous and bounded functions. These coresets preserve $F(x)$ only for $x$ within a specific ball, which means that they do not satisfy the requirements of a strong coreset. This naturally leads to the following question.

**Question 1.1.** *Consider an optimization problem of the form $\min_{x \in \mathbb{R}^d} F(x)$, where $F(x) = F_1(x) + \cdots + F_n(x)$. Under what conditions can an efficient construction of a strong $\varepsilon$-coreset of small size be achieved for the robust version $F^{(m)}(x)$?*

Our answer to Question 1.1 is perhaps surprising. We show that two simple conditions are sufficient for the existence of a small coreset for $F^{(m)}$: $F(x)$ has a small vanilla coreset and has bounded total sensitivity. The first condition is natural, as the robust coreset extends the concept of a vanilla coreset. The second condition, based on the existence of small vanilla coresets, is a mild condition, since a majority of current vanilla coreset constructions rely on bounded total sensitivity. We remark that the total sensitivity is a widely used complexity measure in sparsification problems and has been a crucial quantity in establishing theoretical guarantees for coreset sizes across many classical machine learning problems, including subspace embeddings, regression, PCA, clustering, and projective clustering (Drineas et al., 2006; Woodruff, 2014; Feldman & Langberg, 2011; Braverman et al., 2021a; Woodruff & Yasuda, 2023).

### 1.1. Problem Definition

Suppose that $\mathcal{F} = \{(f, \omega_f)\}$ is a finite weighted set of functions, where each $f : \mathbb{R}^d \to \mathbb{R}_{\geq 0}$ is associated with a weight $\omega_f \geq 0$. The loss function for $\mathcal{F}$ is $\mathcal{L}(\mathcal{F}; x) = \sum_{(f, \omega_f) \in \mathcal{F}} \omega_f \cdot f(x)$ and the robust version of the loss function with $m$ outliers is defined as

$$\mathcal{L}^{(m)}(\mathcal{F}; x) = \min_{\substack{\mathcal{F}' \subseteq \mathcal{F} \\ |\mathcal{F} \setminus \mathcal{F}'| \leq m}} \sum_{(f, \omega_f) \in \mathcal{F}'} \omega_f \cdot f(x).$$

The associated optimization problem is to solve $\min_{x \in \mathbb{R}^d} \mathcal{L}^{(m)}(\mathcal{F}; x)$.

When $\omega_f = 1$ for all $f \in \mathcal{F}$, we also say that $\mathcal{F}$ is unweighted and write $\mathcal{F} = \{f\}$.

**Coresets** Suppose that $\tilde{\mathcal{F}}$ is a weighted (multi-) subset of $\mathcal{F}$; that is, each function in $\tilde{\mathcal{F}}$ is also in $\mathcal{F}$ and each function $f$ in $\tilde{\mathcal{F}}$ is associated with a weight $\omega_f \geq 0$ (for some $f \in \mathcal{F}$, multiple $f$ with different weights may appear in $\tilde{\mathcal{F}}$).

We say $\tilde{\mathcal{F}}$ is an $(\varepsilon, m)$-robust coreset (or simply, an $(\varepsilon, m)$-coreset) of $\mathcal{F}$ if $\tilde{\mathcal{F}}$ is a weighted subset of $\mathcal{F}$ and it holds for every $x \in \mathbb{R}^d$ and every $t = 0, 1, \ldots, m$ that

$$(1 - \varepsilon) \mathcal{L}^{(t)}(\mathcal{F}; x) \leq \mathcal{L}^{(t)}(\tilde{\mathcal{F}}; x) \leq (1 + \varepsilon) \mathcal{L}^{(t)}(\mathcal{F}; x).$$

An $(\varepsilon, 0)$-coreset is also referred to as a (vanilla) $\varepsilon$-coreset.

The typical way to reduce the scale of the problem is to solve, instead of the original $\min_{x \in \mathbb{R}^d} \mathcal{L}^{(m)}(\mathcal{F}; x)$, the coreset version $\min_{x \in \mathbb{R}^d} \mathcal{L}^{(m)}(\tilde{\mathcal{F}}; x)$ if $\tilde{\mathcal{F}}$ is an $(\varepsilon, m)$-coreset of $\mathcal{F}$. The following is a folklore result on the guarantee of the optimal solution of the coreset version.

**Lemma 1.2.** *Suppose that $\tilde{\mathcal{F}}$ is an $(\varepsilon, m)$-robust coreset of $\mathcal{F}$ and $\varepsilon \in (0, 1)$. Let $\hat{x} = \arg \min_{x \in \mathbb{R}^d} \mathcal{L}^{(m)}(\tilde{\mathcal{F}}; x)$ and $x^* = \arg \min_{x \in \mathbb{R}^d} \mathcal{L}^{(m)}(\mathcal{F}; x)$. It holds that $\mathcal{L}^{(m)}(\mathcal{F}; \hat{x}) \leq \frac{1+\varepsilon}{1-\varepsilon} \mathcal{L}^{(m)}(\mathcal{F}; x^*)$.*

Below we describe how two classical problems, computing a subspace embedding and computing a clustering, fit in our framework.

**Subspace Embedding** Let $p \geq 1$. For a matrix $A \in \mathbb{R}^{n \times d}$ ($n \gg d$), we say a matrix $B \in \mathbb{R}^{m \times d}$ with $m \ll n$ is an $\ell_p$-subspace embedding for $A$ with distortion parameter $\varepsilon$ if $(1 - \varepsilon)\|Ax\|_p^p \leq \|Bx\|_p^p \leq (1 + \varepsilon)\|Ax\|_p^p$ holds simultaneously for all $x \in \mathbb{R}^d$. A typical construction of $B$ is sampling rows of $A$ by Lewis weights and rescaling the samples. This can be viewed within our sparsification framework by defining $f_i(x) = |\langle a_i, x \rangle|^p$ and $\mathcal{F} = \{f_1, \ldots, f_n\}$, where $a_i$ denotes the $i$-th row of $A$, so that $\mathcal{L}(\mathcal{F}; x) = \|Ax\|_p^p$. Suppose that $\tilde{\mathcal{F}} = \{(f_{i_j}, w_{i_j})\}$ is an $\varepsilon$-coreset of $\mathcal{F}$, where $i_1, \ldots, i_m \in [n]$ denote the indices of sampled rows, then one can take the $j$-th row of $B$ to be $b_j = a_{i_j} \cdot w_{i_j}^{1/p}$. Then the $\varepsilon$-coreset property gives exactly that $B$ is an $\ell_p$-subspace embedding for $A$ with distortion parameter $\varepsilon$.

**$k$-Median Coresets** Suppose that $X = \{x_1, \ldots, x_n\}$ are $n$ points in $\mathbb{R}^d$. Define a set of $k$ centers $C = \{c_1, \ldots, c_k\}$, where $c_i \in \mathbb{R}^d$. The clustering cost of $X$ with respect to $C$ is defined to be $\sum_{i=1}^n \min_\ell \|x_i - c_\ell\|_2$. The coreset for the $k$-median problem is to find a subset $X' = \{x_{i_1}, \ldots, x_{i_m}\}$ with weights $w_1, \ldots, w_m \geq 0$ such that the weighted clustering cost of $X'$ approximates the clustering cost of $X$ for every choice of $C$; that is, $\sum_{j=1}^m w_j \min_\ell \|x_{i_j} - c_\ell\|_2 = (1 \pm \varepsilon) \sum_{i=1}^n \min_\ell \|x_i - c_\ell\|_2$ for all subsets of $C \subset \mathbb{R}^d$ with $|C| = k$. This again can be viewed within our sparsification framework by defining $f_i(c_1, \ldots, c_k) = \min_j \|x_i - c_j\|_2$ and $\mathcal{F} = \{f_1, \ldots, f_n\}$. An $\varepsilon$-coreset of the $k$-median problem is exactly an $\varepsilon$-coreset of $\mathcal{F}$.

### 1.2. Related Works

Coresets for clustering have been a rich and extensively studied research area for over 20 years. Stemming from (Har-Peled & Mazumdar, 2004; Har-Peled & Kushal, 2007), early coreset algorithms rely on ad-hoc geometric constructions, giving size bounds that are exponential in the dimension $d$. Chen (2009); Langberg & Schulman (2010); Feldman & Langberg (2011) initiated the use of sampling algorithms in constructing coresets for clustering, achieving size bounds

polynomial in $d$. Recently, coresets with size independent of $d$ have been constructed by employing modern dimension reduction techniques for clustering and new group sampling methods (Feldman et al., 2020; Sohler & Woodruff, 2018; Huang & Vishnoi, 2020; Cohen-Addad et al., 2021; 2022; Cohen-Addad et al., 2022; Huang et al., 2024).

Beyond robustness, coresets have also been explored in various other settings to address new challenges in machine learning. For instance, Bachem et al. (2018); Braverman et al. (2019) give simultaneous coresets to handle multiple objectives, Braverman et al. (2021b) design coresets for datasets with missing values, Huang et al. (2021) construct coresets for time series data, and Huang et al. (2019); Bandyapadhyay et al. (2021); Braverman et al. (2022) consider the setting with fairness constraints.

## 2. Preliminaries

**Notation** In this paper, we always assume that the function set is finite. We use $[n]$ to denote the set $\{1, \ldots, n\}$. The notations $f \lesssim g$ and $f \gtrsim g$ indicate $f \leq Cg$ and $f \geq Cg$, respectively, for some constant $C > 0$.

For a matrix $A$, its Frobenius norm is denoted by $\|A\|_F := (\sum_{i,j} a_{ij}^2)^{1/2}$ and its operator norm by $\|A\|_{op} := \sup_{x \neq 0} \|Ax\|_2 / \|x\|_2$.

For a random variable $X$, we write $X \sim \mathcal{D}$ to indicate that $X$ follows the probability distribution $\mathcal{D}$. For a finite set $S$, we denote by $\mathrm{Unif}(S)$ the uniform distribution on $S$. We denote by $\mathrm{Geometric}(p)$ the geometric distribution with expected value $1/p$.

**Sensitivities** For a weighted set $\mathcal{F} = \{(f, w_f)\}$, the sensitivity of $f \in \mathcal{F}$ is defined as

$$\sigma_{\mathcal{F}}(f) = \sup_{x \in \mathbb{R}^d} \frac{w_f f(x)}{\sum_{(g, w_g) \in \mathcal{F}} w_g g(x)}.$$

The sensitivity of $\mathcal{F}$ is defined as

$$\sigma_{\mathcal{F}} = \sum_{f \in \mathcal{F}} \sigma_{\mathcal{F}}(f).$$

We say that $\mathcal{F}$ has total sensitivity $T$ if $\sigma_{\mathcal{F}'} \leq T$ for every $\mathcal{F}' \subseteq \mathcal{F}$.

We note that it may be difficult to compute the precise values of sensitivities $\sigma_{\mathcal{F}}(f)$. In fact, it suffices to compute an upper bound $\tilde{\sigma}_{\mathcal{F}}(f) \geq \sigma_{\mathcal{F}}(f)$ such that $\tilde{\sigma}_{\mathcal{F}} = \sum_{f \in \mathcal{F}} \tilde{\sigma}_{\mathcal{F}}(f) = O(\sigma_{\mathcal{F}})$. To simplify the presentation, we also refer to $\tilde{\sigma}_{\mathcal{F}}(f)$ as sensitivities.

For $\ell_2$-subspace embeddings, the sensitivity is a classic quantity called the leverage score. Constant-factor approximation of the leverage scores of a matrix $A \in \mathbb{R}^{n \times d}$ can be found in time $O(nd + \mathrm{poly}(d))$ (see, e.g., (Drineas et al., 2006; Woodruff, 2014)).

## 3. Our Results and Technical Overview

Our main result is the following robust coreset construction algorithm.

**Theorem 3.1** (Informal statement of Theorem 4.1)**.** *There exists an algorithm, for every loss function $F(x) = \sum_{f \in \mathcal{F}} f(x)$ such that $\mathcal{F}$ has total sensitivity $T$ and admits a vanilla coreset of size $Q$, constructs with probability at least 0.99 an $(\varepsilon, m)$-coreset for $\mathcal{F}$ with size $O(\frac{Tm}{\varepsilon} \cdot \log \frac{Tm}{\varepsilon}) + Q$.*

When applied to $\ell_2$ subspace embeddings, this theorem gives a coreset size of $\tilde{O}(\frac{md}{\varepsilon} \log \frac{md}{\varepsilon}) + Q$. We show in Section D that a size of $\Omega(\frac{md}{\varepsilon})$ is necessary for an $(\varepsilon, m)$-coreset. Note that an $(\varepsilon, m)$-coreset must also be a vanilla $\varepsilon$-coreset, which shows that $Q$ is also necessary. Thus our bound is nearly tight for $\ell_2$ subspace embeddings.

Building upon our coresets, we develop new algorithms for robust optimization tasks. For example, we design algorithms for robust regression (Theorem 5.1) and robust PCA (Theorem 5.3) with a runtime of $d^{O(m)} e^{O(m/\varepsilon)} + O(nd)$, using a sensitivity flattening technique. Additionally, our coresets can be used to improve existing algorithms, such as (Simonov et al., 2019). Furthermore, Theorem 3.1 yields a new coreset for robust $k$-median. Detailed discussions can be found in Section 5. We remark that our coreset construction also extends to another popular setting that removes a total weight of $m$ instead of exactly $m$ functions of $\mathcal{F}$; see Appendix C.

In Section 6, we conduct experiments on real-word datasets to demonstrate that our coreset constructions are effective in approximating the loss function and considerably reduce the running time for robust regression problems while maintaining a good approximation of the objective function.

### 3.1. Technical Overview

Our coreset construction algorithm contains two stages. In the first stage, we identify a small set $S \subset \mathcal{F}$ containing "contributing" functions and include $S$ in the coreset with unit weights. In the second stage, we compute a vanilla coreset for $\mathcal{F} \setminus S$ and apply a refinement process (Algorithm 2) to adapt it for robust objectives.

We begin with the construction of $S$. A function $f \in A$ is called *contributing* if for some $x \in \mathbb{R}^d$, $f(x) \geq \frac{\varepsilon}{m} \cdot \mathcal{L}^{(m)}(A; x)$. Our goal is to include all contributing functions in $S$, since this will ensure that removing any $m$ functions from $\mathcal{F} \setminus S$ incurs at most $m \cdot \frac{\varepsilon}{m} = \varepsilon$ relative error. To do this, we sample each $f \in \mathcal{F}$ with probability $\frac{1}{m}$ and then include in $S$ the functions of $\Omega(\varepsilon)$ sensitivity within the sample. We can repeat this procedure sufficiently many times to ensure that all contributing functions are included. We need to (i) show for one round that each contributing $f$ will, with a good probability, survive the sampling and have

$\Omega(\varepsilon)$ sensitivity within the sample; and (ii) bound the total number of contributing functions.

We first establish (i). Consider an event $\mathcal{E}$ that $f$ survives the sampling while all outliers of $\mathcal{L}^{(m)}(A; x)$ except $f$ do not. Then $\Pr(\mathcal{E}) \geq (1 - \frac{1}{m})^m \cdot \frac{1}{m} = \Omega(\frac{1}{m})$. By Markov's inequality, we know that conditioned on $\mathcal{E}$, with constant probability, the aggregation of the sample on $x$ is at most $O(\frac{1}{m}) \cdot \mathcal{L}^{(m)}(A; x)$, so the sensitivity of $f$ within the sample is at least $\frac{\varepsilon}{m} / O(\frac{1}{m}) = \Omega(\varepsilon)$. We have therefore established (i) with the "good probability" being $\Omega(\frac{1}{m})$.

Next we examine (ii). Since the total sensitivity of $\mathcal{F}$ is at most $T$, there are $O(\frac{T}{\varepsilon})$ functions with sensitivity at least $\Omega(\varepsilon)$, which implies that each round returns $O(\frac{T}{\varepsilon})$ functions. If the number of contributing functions is $N$, then by a union bound, we need $O(m \log N)$ rounds of sampling to ensure with constant probability that all contributing functions are included. This implies that $N \leq |S| = O(\frac{Tm}{\varepsilon} \log N)$. Solving this inequality for $N$ gives that $N = O(\frac{Tm}{\varepsilon} \log \frac{Tm}{\varepsilon})$.

Now, consider the second stage. We begin by constructing a vanilla $\varepsilon$-coreset $D$ of $\mathcal{F} \setminus S$, where $|D| = Q$. A potential issue is that the weight of a function in $D$ may be too large and its removal could result in a violation of the coreset property. To resolve this, we "split" functions in $D$ into multiple copies, each with a smaller weight. Specifically, we split each $f$ into $\lceil \frac{m}{\varepsilon} \cdot \sigma_D(f) \rceil$ functions. The resulting size of the modified coreset is at most $\sum_{f \in D} \frac{m}{\varepsilon} \cdot \sigma_D(f) + 1 \leq \frac{Tm}{\varepsilon} + Q$, which is still affordable. Moreover, by the property of sensitivity and the guarantee of the first stage that $S$ contains all contributing functions, we know that removing any $m$ split functions will incur a total error at most $\frac{m}{m/\varepsilon} \cdot \mathcal{L}(D; x) \leq O(\varepsilon) \cdot \mathcal{L}(\mathcal{F} \setminus S; x) \leq O(\varepsilon) \cdot \mathcal{L}^{(m)}(\mathcal{F}; x)$.

# 4. Coreset Construction

Our main result is the following theorem.

**Theorem 4.1.** *Consider a sparsification problem for $F(x) = \sum_{f \in \mathcal{F}} f(x)$ and $\varepsilon \in (0, \frac{1}{2})$. Suppose that $\mathcal{F}$ has total sensitivity $T$ and there exists an algorithm that computes a vanilla $\varepsilon$-coreset for $F$ of size $Q$. Then, Algorithm 3 computes an $(\varepsilon, m)$-robust coreset for $F$ of size $O(\frac{Tm}{\varepsilon} \cdot \log \frac{Tm}{\varepsilon}) + Q$, with probability at least $0.99$. Moreover, if the vanilla coreset algorithm runs in time $t_0(n, \varepsilon)$ on $n$ input points and the sensitivity oracle computes the sensitivities of $n$ points in time $t_1(n)$, then Algorithm 3, with probability at least $0.99$, runs in time $O(t_0(|\mathcal{F}|, \varepsilon)) + t_1(\frac{n}{m} + O(\sqrt{\frac{n}{m} \log \frac{mT}{\varepsilon}})) \cdot m \log(\frac{mT}{\varepsilon})$.*

---

**Algorithm 1** $\mathrm{Uniform}(A, \varepsilon, m)$

---

**Input:** A set $A$ of functions, parameters $\varepsilon$ and $m$
**Output:** A subset $D \subseteq A$
1:  $B \leftarrow \emptyset$
2:  for each $f \in A$, with probability $\frac{1}{m}$, add $f$ to $B$
3:  for each $f \in B$, compute the sensitivity $\sigma_B(f)$
4:  $D \leftarrow \{f \in B : \sigma_B(f) \geq \frac{\varepsilon}{4}\}$
5:  **Return** $D$

---

**Algorithm 2** $\mathrm{Refine}(D, \varepsilon, m)$

---

**Input:** A coreset $D$, parameters $\varepsilon$ and $m$
**Output:** A refined subset $\tilde{D}$ adapted for the robust optimization problem
1:  $\tilde{D} \leftarrow \emptyset$
2:  **for** $(f, \omega_f) \in D$ **do**
3:      compute the sensitivity $\sigma_D(f)$
4:      $n_f \leftarrow \lceil \frac{m}{\varepsilon} \cdot \sigma_D(f) \rceil$
5:      Add $n_f$ copies of $(f, \frac{\omega_f}{n_f})$ to $\tilde{D}$
6:  **end for**
7:  **Return** $\tilde{D}$

---

## 4.1. Analysis of Algorithm 3

Let $A$ be a set of functions $f : \mathbb{R}^d \to \mathbb{R}_{\geq 0}$. We need the following definition in our analysis.

**Definition 4.2.** *A function $f \in A$ is called* contributing *if there exists $x \in \mathbb{R}^d$ such that $f(x) \geq \frac{\varepsilon}{m} \cdot \mathcal{L}^{(m)}(A; x)$.*

The following lemma shows that a fixed contributing function will be added to $D$ with probability $\Theta(1/m)$ in each repetition of Algorithm 1.

**Lemma 4.3.** *Assume that $f$ is contributing, then with probability at least $\frac{1}{5m}$, the set returned by $\mathrm{Uniform}(A, \varepsilon, m)$ contains $f$.*

*Proof.* By definition, there exists $x \in \mathbb{R}^d$ such that $f(x) \geq \frac{\varepsilon}{m} \cdot \mathcal{L}^{(m)}(A; x)$. Fix this $x$. Let $L \subset A$ denote the set of outliers excluded by $\mathcal{L}^{(m)}(A; x)$; namely, $L$ consists of the $m$ functions $f$ in $A$ with largest values of $f(x)$.

Let $L' = L \setminus \{f\}$ so $|L'| \leq m$. Consider the event $\mathcal{E}$ where $f \in B$ while $L' \cap B = \emptyset$. Then $\Pr[\mathcal{E}] = (1 - \frac{1}{m})^{|L'|} \cdot \frac{1}{m} \leq \frac{1}{em}$. Observe that

$$\mathbb{E}\left[\sum_{h \in B} h(x) \,\Big|\, \mathcal{E}\right] = f(x) + \frac{1}{m} \sum_{h \in A \setminus (L \cup \{f\})} hh(x)$$

$$\leq f(x) + \frac{1}{m} \cdot F^{(m)}(x)$$

$$\leq (1 + \frac{1}{\varepsilon}) \cdot f(x)$$

$$\leq \frac{2}{\varepsilon} \cdot f(x).$$

**Algorithm 3** Coreset$(A, \varepsilon, m)$

**Input:** A set $A$ of functions, parameters $\varepsilon$ and $m$, and an algorithm Vanilla$(A)$ to construct an $\varepsilon$-coreset for $A$

**Output:** An $(\varepsilon, m)$-robust coreset for $A$

1: $S \leftarrow \emptyset$
2: $R \leftarrow \Theta(m \log \frac{Tm}{\varepsilon})$
3: **for** $i = 1, 2, \cdots, R$ **do**
4:     $D \leftarrow \text{Uniform}(A, \varepsilon, m)$
5:     $S \leftarrow S \cup D$
6: **end for**
7: $V \leftarrow \text{Vanilla}(A \setminus S)$
8: $\tilde{S} \leftarrow \{(f, 1) : f \in S\}$
9: **Return** $\tilde{S} \cup \text{Refine}(V, \varepsilon, m)$.

---

By Markov's inequality, with probability at least $\frac{1}{2} \cdot \frac{1}{em} \geq \frac{1}{5m}$, the event $\mathcal{E}$ happens and $\sum_{h \in B} h(x) \leq \frac{4}{\varepsilon} \cdot f(x)$. This implies that $\sigma_B(f) \geq \frac{f(x)}{\sum_{h \in B} h(x)} \geq \frac{\varepsilon}{4}$. Therefore, $f$ is added to $D$ with probability at least $\frac{1}{5m}$. $\qquad \square$

**Lemma 4.4.** *The number of contributing functions in $A$ is $O(\frac{Tm}{\varepsilon} \cdot \log \frac{Tm}{\varepsilon})$.*

*Proof.* Define a sequence $a_0 = n$, $a_i = \frac{20Tm}{\varepsilon} \cdot \log(2a_{i-1})$ for $i \geq 1$. We prove by induction that the number of contributing functions is upper bounded by $a_i$ for every $i \geq 0$.

The statement is trivial for $i = 0$, since the total number of functions is always bounded by $n$. Now, assume the number of contributing functions is bounded by $a_i$, we prove that the number is also bounded by

$$a_{i+1} = \frac{20Tm}{\varepsilon} \cdot \log(2a_i). \tag{1}$$

To see this, we remark that for an contributing $f \in A$, a single repetition of Algorithm 1 returns $f$ with probability at least $\frac{1}{5m}$. Now consider $5m \log(2a_i)$ independent repetitions of $\text{Uniform}(F, \varepsilon, m)$. By a union bound, we see that with probability

$$1 - a_i \cdot (1 - \frac{1}{5m})^{5m \log(2a_i)} \geq 1 - a_i \cdot \frac{1}{2a_i} = \frac{1}{2},$$

all contributing functions will be found. Since the total sensitivity is bounded by $T$, there are at most $\frac{4T}{\varepsilon}$ functions added into $D$ in each repetition, which implies that the number of total contributing functions is bounded by $5m \log(2a_i) \cdot \frac{4T}{\varepsilon} = \frac{20Tm}{\varepsilon} \cdot \log(2a_i)$. This establishes (1).

We may continue iterating as long as $a_{i+1} \leq a_i$; otherwise, we have $\frac{a_i}{\log(2a_i)} \leq \frac{20Tm}{\varepsilon}$ and so $a_i = O(\frac{Tm}{\varepsilon} \log \frac{Tm}{\varepsilon})$ as desired. If the process does not terminate, the sequence $\{a_i\}$ is a monotone decreasing and positive sequence, thus converging to a unique limit $\lim_{i \to \infty} a_i = a$. Letting $i \to \infty$ on both sides of (1), we obtain that $\frac{a}{\log(2a)} = \frac{20Tm}{\varepsilon}$, which also implies $a = O(\frac{Tm}{\varepsilon} \log \frac{Tm}{\varepsilon})$, as desired. $\qquad \square$

We are ready to prove Theorem 4.1.

*Proof of Theorem 4.1.* Let $P$ denote the return of Coreset$(A, \varepsilon, m)$. We first bound $|P|$. Since the total sensitivity of $F$ is at most $T$, at most $\frac{4T}{\varepsilon}$ functions in $A$ are added into $S$ in each repetition of Algorithm 1. Thus, $|S| = O(\frac{Tm}{\varepsilon} \log \frac{Tm}{\varepsilon})$. It remains to bound the set returned by Algorithm 2. Observe that

$$|\tilde{D}| \leq \sum_{(f, \omega_f) \in D} n_f = \sum_{(f, \omega_f) \in D} \left\lceil \frac{m}{\varepsilon} \cdot \sigma_D(f) \right\rceil$$

$$\leq \sum_{(f, \omega_f) \in D} \left( \frac{m}{\varepsilon} \cdot \sigma_D(f) + 1 \right) \leq \frac{Tm}{\varepsilon} + Q.$$

The coreset size is hence bounded by $O(\frac{Tm}{\varepsilon} \log \frac{Tm}{\varepsilon}) + Q$.

Next, we prove that $P$ is indeed an $(\varepsilon, m)$-coreset. We shall show that for every $x \in \mathbb{R}^d$ and $t = 0, \ldots, m$, $\mathcal{L}^{(t)}(P; x) \in (1 \pm C\varepsilon) \cdot \mathcal{L}^{(t)}(F; x)$ for some absolute constant $C$. The proof will be completed by rescaling $\varepsilon$.

We first prove that $\mathcal{L}^{(t)}(P; x) \leq (1 + C\varepsilon) \mathcal{L}^{(t)}(A; x)$. It suffices to remove at most $t$ items from $P$ and show that the weighted sum of the remaining items is at most $(1 + C\varepsilon) \mathcal{L}^{(t)}(A; x)$.

Let $L_x$ denote the set of outliers excluded by $\mathcal{L}^{(t)}(P; x)$, then $|L_x| \leq t \leq m$. By Lemma 4.3 and Lemma 4.4, we know that with constant probability, every contributing $F_j$ has been added into $S$. Conditioning on this event,

$$\sum_{f \in L_x \setminus S} f(x) \leq |L_x| \cdot \frac{\varepsilon}{m} \cdot \mathcal{L}^{(m)}(A; x) \leq \varepsilon \cdot \mathcal{L}^{(m)}(P; x)$$

$$\leq \varepsilon \cdot \mathcal{L}^{(t)}(A; x).$$

Let $L'_x = \{(f, 1) \mid f \in L_x \cap S\}$ to be removed from $P$. Clearly $|L'_x| \leq t$. Since $L'_x \subseteq S$ and $\text{Refine}(V, F, \varepsilon, m)$ returns an $\varepsilon$-coreset for $A \setminus S$, we have

$$\sum_{(f, \omega_f) \in P \setminus L'_x} \omega_f \cdot f(x)$$

$$\leq \sum_{g \in S \setminus L_x} g(x) + (1 + \varepsilon) \cdot \sum_{g \in A \setminus S} g(x)$$

$$\leq (1 + \varepsilon) \sum_{g \in A \setminus (L_x \cap S)} g(x)$$

$$\leq (1 + \varepsilon) \left( \sum_{g \in A \setminus L_x} g(x) + \sum_{g \in L_x \setminus L'_x} g(x) \right)$$

$$\leq (1 + 2\varepsilon) \mathcal{L}^{(t)}(A; x).$$

We have estalished that $\mathcal{L}^{(t)}(P; x) \leq (1 + C\varepsilon) \mathcal{L}^{(t)}(A; x)$.

It remains to prove that $\mathcal{L}^{(t)}(P; x) \geq (1 - C\varepsilon) \mathcal{L}^{(t)}(A; x)$. It suffices to ensure that, after removing at most $t$ items

from $A$, the sum of the remaining items is at most $(1 + C\varepsilon)\, \mathcal{L}^{(t)}(P; x)$ (where $C$ could be different).

Let $G_x$ denote the set of outliers excluded by $\mathcal{L}^{(t)}(P; x)$. Let $\tilde{G}_x = \{f \mid (f, 1) \in \tilde{S} \cap G_x\}$ to be removed from $A$. Clearly $|\tilde{G}_x| \leq t$. We shall show that

$$\sum_{f \in A \setminus \tilde{G}_x} f(x) \leq (1 + C\varepsilon)\, \mathcal{L}^{(t)}(P; x).$$

Note that

$$
\begin{aligned}
\mathcal{L}^{(t)}(P; x) &= \sum_{(f,1) \in \tilde{S} \setminus \tilde{G}_x} f(x) \;+\; \sum_{f \in P \setminus (\tilde{G}_x \cup \tilde{S})} \omega_f \cdot f(x) \\
&= \sum_{f \in S \setminus G_x} f(x) \;+\; \sum_{(f,\omega_f) \in P \setminus \tilde{S}} \omega_f \cdot f(x) - \sum_{(f,\omega_f) \in G_x \setminus \tilde{S}} \omega_f \cdot f(x) \\
&\geq \sum_{f \in S \setminus G_x} f(x) + \frac{1}{1-\varepsilon} \sum_{f \in A \setminus S} f(x) - \sum_{(f,\omega_f) \in G_x \setminus \tilde{S}} \omega_f \cdot f(x),
\end{aligned}
$$

where we used the fact that $P \setminus S$ is an $\varepsilon$-coreset of $A \setminus S$. We need to upper bound $\sum_{(f,\omega_f) \in G_x \setminus \tilde{S}} \omega_f \cdot f(x)$. Suppose that there are $n'_f$ copies of the same $f$ in $G_x \setminus \tilde{S}$. Then

$$
\begin{aligned}
\sum_{(f,\omega_f) \in G_x \setminus \tilde{S}} \omega_f \cdot f(x) &= \sum_{\substack{f \in G_x \setminus S \\ (f,\omega_f) \in V}} \frac{\omega_f n'_f}{\lceil \frac{m}{\varepsilon} \sigma_{A \setminus S}(f) \rceil} f(x) \\
&\leq \sum_{\substack{f \in G_x \setminus S \\ (f,\omega_f) \in V}} \frac{n'_f \varepsilon}{m} \cdot \frac{\omega_f \cdot f(x)}{\sigma_{A \setminus S}(f)} \\
&\leq \frac{\varepsilon}{m} \cdot \sum_{\substack{f \in G_x \setminus S \\ (f,\omega_f) \in V}} n'_f \cdot \sum_{(\omega_g, g) \in V} \omega_g \cdot g(x) \\
&\leq \varepsilon \cdot (1 + \varepsilon) \sum_{f \in A \setminus S} f(x).
\end{aligned}
$$

It follows that

$$
\begin{aligned}
\mathcal{L}^{(t)}(P; x) &\geq \sum_{f \in S \setminus G_x} f(x) + \left( \frac{1}{1-\varepsilon} - \varepsilon(1+\varepsilon) \right) \sum_{f \in A \setminus S} f(x) \\
&\geq \sum_{f \in S \setminus G_x} f(x) + \sum_{f \in A \setminus S} f(x) \geq \sum_{f \in A \setminus \tilde{G}_x} f(x).
\end{aligned}
$$

We have completed the proof of the correctness of the coreset. Next we analyse the runtime.

We begin by analyzing Algorithm 1. Line 1 can be implemented by indexing functions $f \in A$ as $1, 2, \ldots, n = |A|$, generating independent Geometric$(1/m)$ variables $Z_1, Z_2, \ldots$ and selecting functions at indices $Z_1, Z_1 + Z_2$, $Z_1 + Z_2 + Z_3$, and so on. Thus, Line 1 takes $O(|B|)$ time, assuming $O(1)$ time to generate a geometric random variable. Lines 1 and 1 take $t_1(|B|)$ and $O(|B|)$ time, respectively.

Hence, the overall runtime of Algorithm 1 is $O(t_1(|B|))$. By a Chernoff bound, we see that $|B| \leq \frac{n}{m} + O(\sqrt{\frac{n}{m} \log \frac{1}{\delta}})$ with probability at least $1 - \delta$.

Algorithm 2 runs in time $O(\sum_{(f, \omega_f) \in D} n_f + |D|) = O(mT/\varepsilon + |D|)$.

Back to Algorithm 3. By the analysis above, with probability at least 0.99, the invocation of Uniform in every iteration of the for-loop generates a set $B$ of size $|B| \leq \frac{n}{m} + O(\sqrt{\frac{n}{m} \log R})$. In total, the for-loop runs in time $R \cdot t_1(\frac{n}{m} + O(\sqrt{\frac{n}{m} \log R}))$. Line 3 runs in time $t_0(n, \varepsilon)$, Line 3 in time $O(n) = O(t_0(n, \varepsilon))$ and Line 3 in time $O(mT/\varepsilon + n)$. The overall runtime is therefore $O(t_0(n, \varepsilon)) + R \cdot t_1(\frac{n}{m} + O(\sqrt{\frac{n}{m} \log R}))$. $\qquad \square$

## 5. Applications

**Robust Regression** The standard $\ell_2$-regression problem is to solve $\min_{x \in \mathbb{R}^d} \|Ax - b\|_2$, for which a standard approach to reduce the dimension is to solve instead $\min_{x \in \mathbb{R}^d} \|SAx - Sb\|_2$, where $S$ is an $\ell_2$-subspace embedding matrix for the concatenated matrix $\begin{pmatrix} A & b \end{pmatrix}$. Regarding $\ell_2$-subspace embeddings, the total sensitivity $T = d$ and the vanilla coreset size can be made $Q = O(\frac{d}{\varepsilon^2})$ using a poly$(n)$ time algorithm (Batson et al., 2012) or $Q = O(\frac{d \log d}{\varepsilon^2})$ using classical leverage score sampling that runs in time $O(nd + \text{poly}(d))$ (see, e.g., (Drineas et al., 2006; Woodruff, 2014)). In the interest of runtime, we shall use leverage score sampling and thus, by Theorem 4.1, the $(\varepsilon, m)$-coreset size is $O(\frac{md}{\varepsilon} \cdot \log \frac{md}{\varepsilon} + \frac{d \log d}{\varepsilon^2})$. Based on our coreset construction, we present Algorithm 4 for robust $\ell_2$-regression with $m$ outliers, which is to solve $\min_{x \in \mathbb{R}^d} F^{(m)}(Ax - b)$, where $F^{(m)}(u)$ denotes the sum of $u_i^2$ except the $m$ largest coordinates (in absolute values).

**Theorem 5.1.** *Suppose that $\varepsilon \in (0, \frac{1}{4})$. Algorithm 4 returns an $\tilde{x}$ which satisfies with probability at least $0.9$ that*

$$F^{(m)}(A\tilde{x} - b) \leq (1 + 14\varepsilon) \min_{x \in \mathbb{R}^d} F^{(m)}(Ax - b)$$

*The algorithm runs in time $d^{O(m)} e^{O(m/\varepsilon)} + O(nd)$.*

We need the following lemma, whose proof is deferred to Appendix A.

**Lemma 5.2.** *Suppose that $A \in \mathbb{R}^{n \times d}$ and $b \in \mathbb{R}^n$, where the leverage scores of $A$ are bounded by $1/r$. Let $S \in \mathbb{R}^{n \times n}$ be a random diagonal matrix, where $S_{ii} = \sqrt{2}$ with probability $1/2$ and $S_{ii} = 0$ with probability $1/2$. Let $x^* = \arg\min_{x \in \mathbb{R}^n} \|Ax - b\|_2$ and $\tilde{x} = \arg\min_{x \in \mathbb{R}^n} \|SAx - Sb\|_2$. When $r \gtrsim d \log d + 1/\varepsilon$, it holds with probability at least $0.98$ that $\|A\tilde{x} - b\|_2 \leq (1 + \varepsilon)\|Ax^* - b\|_2$.*

Now we are ready to prove Theorem 5.1.

*Proof of Theorem 5.1.* Let $x^* = \arg\min_{x \in \mathbb{R}^d} F^{(m)}(Ax - $

**Algorithm 4** RobustRegression$(A, b, \varepsilon, m)$

**Input:** $A \in \mathbb{R}^{n \times d}$ and $b \in \mathbb{R}^n$, parameters $\varepsilon$ and $m$
**Output:** $(1 + \varepsilon)$-approx. solution to $\min_x F^{(m)}(Ax - b)$
1: $(D, \{w_i\}) \leftarrow \text{Coreset}((A \; b), \varepsilon, m)$ {Each row $D_{i,*} \in \mathbb{R}^{d+1}$ is associated with weight $w_i$}
2: Rescale each row of $D$ as $D_{i,*} \leftarrow \sqrt{w_i} D_{i,*}$
3: $r \leftarrow O(\log d + 1/\varepsilon)$
4: Duplicate each row of $D$ by $r$ times and rescale by $\frac{1}{\sqrt{r}}$, yielding $D' = (H' \; y')$ {$H' \in \mathbb{R}^{rn \times d}$ and $y' \in \mathbb{R}^{rn}$}
5: $L \leftarrow 2^{O(rm)}$
6: $X \leftarrow \emptyset$
7: **for** $i = 1, 2, \ldots, L$ **do**
8:     Generate a diagonal matrix $S$ with i.i.d. diagonal entries $S_{ii} \sim \text{Unif}(\{0, \sqrt{2}\})$
9:     Compute $x' = \arg\min_{x \in \mathbb{R}^d} \|SH'x - Sy'\|_2$
10:     $X \leftarrow X \cup \{x'\}$
11: **end for**
12: Compute $\tilde{x} \leftarrow \arg\min_{x \in X} F^{(rm)} \|H'x - y'\|_2$
13: **Return** $\tilde{x}$

$b)$ and $D = (H \; y)$ (after executing Line 2). Let $J$ denote the set of the indices of the largest $m$ coordinates of $Hx^\sharp - y$, where $x^\sharp = \arg\min_{x \in \mathbb{R}^d} F^{(m)}(Hx - y)$, and $J'$ denote the $rm$ indices in $H'x^\sharp - y'$ that correspond to the indices in $J$.

Consider the event $\mathcal{E}$ that $SH'$ contains none of the rows of indices in $J$. This event happens with probability $2^{-rm}$ in one trial of Line 4, hence it will happen in at least one trial among $L = 2^{O(rm)}$ trials with probability at least 0.99.

Consider the trial in which $\mathcal{E}$ happens. In this case, let $(H'' \; y'')$ be the rows of $(H' \; y')$ whose indices are outside $J'$. It is clear that the leverage scores of $H''$ are at most $1/r$. By Lemma 5.2 and our choice of $r$, we have with probability at least 0.98 that

$$\|H''x' - y''\|_2^2 \leq (1 + \varepsilon)^2 \min_{x \in \mathbb{R}^d} \|H''x - y''\|_2^2$$
$$= (1 + \varepsilon)^2 \|H''x^\sharp - y''\|_2^2 = (1 + \varepsilon)^2 F^{(rm)}(H'x^\sharp - y').$$

It follows that

$$F^{(m)}(H\tilde{x} - y) = F^{(rm)}(H'\tilde{x} - y') \leq F^{(rm)}(H'x' - y')$$
$$\leq \|H''x' - y''\|_2^2 \leq (1 + \varepsilon)^2 F^{(rm)}(H'x^\sharp - y')$$
$$= (1 + \varepsilon)^2 F^{(m)}(Hx^\sharp - y).$$

Using the coreset properties and Lemma 1.2, we then have

$$F^{(m)}(A\tilde{x} - b) \leq \frac{1}{1 - \varepsilon} F^{(m)}(H\tilde{x} - y)$$
$$\leq \frac{(1 + \varepsilon)^2}{1 - \varepsilon} F^{(m)}(Hx^\sharp - y)$$
$$\leq \frac{(1 + \varepsilon)^3}{1 - \varepsilon} F^{(m)}(Ax^\sharp - b)$$

$$\leq \frac{(1 + \varepsilon)^3}{1 - \varepsilon} \cdot \frac{1 + \varepsilon}{1 - \varepsilon} F^{(m)}(Ax^* - b)$$
$$\leq (1 + 14\varepsilon) F^{(m)}(Ax^* - b).$$

This completes the proof of correctness. The overall failure probability is $0.01 + 0.02 = 0.03$, which come from the event $\mathcal{E}$ and Lemma 5.2.

By Theorem 4.1, coreset $D$ has size $N = \tilde{O}(\varepsilon^{-1}md) + O(\varepsilon^{-2}d\log d)$ and thus $D'$ has size $rN$. Each iteration of the for-loop from Line 4 to Line 4 takes time $O((rN)^3) = O(\text{poly}(md/\varepsilon))$. The whole for-loop takes the time $L \cdot \text{poly}(md/\varepsilon) = d^{O(m)}2^{O(m/\varepsilon)}$, which dominates the runtime after Line 1. For Line 1, we have $t_0(n, \varepsilon) = O(nd + \text{poly}(d/\varepsilon))$ and $t_1(n) = O(nd + \text{poly}(d))$ and thus Line 1 takes time $O(nd + n\log\frac{md}{\varepsilon} + \text{poly}(md/\varepsilon))$ with probability at least 0.99. The claimed overall runtime follows. $\square$

**Robust PCA** Given $A \in \mathbb{R}^{n \times d}$, the rank-$k$ PCA of $A$ is given by $AUU^\top$, where $U \in \mathbb{R}^{d \times k}$ has the top-$k$ right singular vectors of $A$ as columns. This $U$ can also be viewed as the minimizer to $\min_{U \in \mathcal{U}} \|A - AUU^\top\|_F$, where $\mathcal{U} = \{U \in \mathbb{R}^{n \times k} : U \text{ has orthonormal columns}\}$. In the robust setting, we let $F^{(m)}(X)$ to denote the sum of $\|x_i\|_2^2$ (where $x_i$ denotes the $i$-th row of $X$) except the $m$ largest rows. The task is to solve $\min_{U \in \mathcal{U}} F^{(m)}(A - AUU^\top)$. Analogously to Theorem 5.1, we have

**Theorem 5.3.** *Suppose that $\varepsilon \in (0, \frac{1}{4})$. There is an algorithm with runtime $d^{O(m)}e^{O(m/\varepsilon)} + O(nd)$ which outputs an $\tilde{U}$ satisfying with probability at least $0.9$ that*

$$F^{(m)}(A\tilde{U}\tilde{U}^\top - A) \leq (1 + 14\varepsilon) \min_{U \in \mathcal{U}} F^{(m)}(AUU^\top - A).$$

The proof is highly similar to that of Theorem 5.1 and is therefore postponed to Appendix B.

A popular application of coreset is to be employed as a pre-processing subroutine to accelerate the existing algorithm. For example, Simonov et al. (2019) give an $n^{O(d^2)}$ time for PCA with $m$ outliers for any $m$. For fixed $m$, we can first run Algorithm 3 to construct an $(\varepsilon, m)$-coreset $S$ for robust PCA. So $|S| = \tilde{O}(dm\varepsilon^{-1} + d\varepsilon^{-2})$. Then we run the algorithm of (Simonov et al., 2019) on $S$ to obtain a $(1 + \varepsilon)$-approximation with running time $(dm\varepsilon^{-1})^{O(d^2)} + \tilde{O}(nd\log(m\varepsilon^{-1}))$.

We remark that Simonov et al. (2019) also prove a lower bound that rules out any constant-factor approximation in time $f(d)n^{o(d)}$, assuming the ETH. At first glance, this appears to contradict the two algorithms given above. However, this lower bound assumes that $m$ can be as large as $\Theta(n)$, in which case our algorithms would also run in time $d^n$, consistent with the lower bound.

**Robust $k$-median** For the $k$-median problem, the total sensitivity is $O(k)$ (Langberg & Schulman, 2010; Feldman &

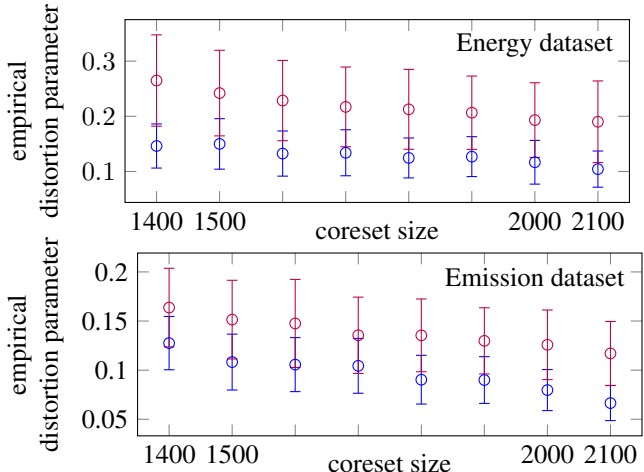

Figure 1: Results of subspace embedding coresets with $m = 10$ and $\varepsilon = 0.25$. Blue plots correspond to our coreset algorithm and the red plots to uniform sampling.

Table 1: Runtimes (in seconds) for robust regression on our coresets and the whole dataset, with $m = 10$, for the Energy dataset (top) and the Emission dataset (bottom).

| Coreset size | Mean | Standard deviation |
|---|---|---|
| 1400 | 1.542 | 0.207 |
| 1500 | 1.602 | 0.214 |
| 1600 | 1.625 | 0.237 |
| 1700 | 1.673 | 0.250 |
| 1800 | 1.699 | 0.225 |
| 1900 | 1.781 | 0.301 |
| 2000 | 1.805 | 0.307 |
| 2100 | 1.860 | 0.317 |
| 19734 (whole dataset) | 13.981 | 0.249 |

| Coreset size | Mean | Standard deviation |
|---|---|---|
| 650 | 0.280 | 0.025 |
| 700 | 0.284 | 0.024 |
| 750 | 0.289 | 0.024 |
| 800 | 0.299 | 0.023 |
| 850 | 0.303 | 0.023 |
| 900 | 0.315 | 0.035 |
| 950 | 0.352 | 0.043 |
| 1000 | 0.396 | 0.022 |
| 36733 (whole dataset) | 7.543 | 0.054 |

Langberg, 2011) and $Q = O(\frac{k}{\varepsilon^2} \cdot \min(k^{1/3}, \varepsilon^{-1}))$ (Cohen-Addad et al., 2022; Huang et al., 2024). Hence, the coreset size is $O\left(\frac{mk}{\varepsilon} \cdot \log \frac{mk}{\varepsilon} + \frac{k}{\varepsilon^2} \cdot \min(k^{1/3}, \varepsilon^{-1})\right)$. In Section C, we compare our results with existing robust coresets for clustering (Huang et al., 2023a; 2025).

# 6. Experiments

We conduct the experiment on two real-world datasets from the UCI Machine Learning Repository: Appliances Energy Prediction[1] (referred to as *Energy*) and Gas Turbine Emission[2] (*Emission*). The Energy dataset has dimension $19735 \times 28$ and the Emission dataset $36733 \times 11$.

**Coreset Verification** We verify that Algorithm 3 produces an effective coreset for subspace embedding with $p = 2$.

Recall that for a matrix $A \in \mathbb{R}^{n \times d}$, the function set $\mathcal{F}$ consists $f_i(x) = |\langle a_i, x \rangle|^2$, where $a_i$ is the $i$-th row of $A$. Fixing parameters $\varepsilon$ and $m$, we indepedently run Algorithm 3 1000 times. Each run produces a coreset $\mathcal{D}_j$ and we compute its empirical distortion parameter $\tilde{\varepsilon}_j = \max_{x \in X} |\mathcal{L}^{(m)}(\mathcal{D}_j; x) / \mathcal{L}^{(m)}(\mathcal{F}; x) - 1|$, where $X \subset \mathbb{R}^d$ consists of 5000 samples drawn independently from $N(0, I_d)$. The size of $\mathcal{D}_j$ is random and is rounded up to $n_j$, the smallest integer multiple of 100, and report $\tilde{\varepsilon}_j$ as the distortion parameter for the specified coreset size $n_j$ (we could pad $\mathcal{D}_j$ with arbitrary unselected $f \in \mathcal{F}$ to achieve size $n_j$, which would only reduce distortion). Finally, we plot the mean and standard deviation of the distortion parameter for

---

[1] https://archive.ics.uci.edu/dataset/374/appliances+energy+prediction
[2] https://archive.ics.uci.edu/dataset/551/gas+turbine+co+and+nox+emission+data+set

different coreset sizes using all $(n_j, \tilde{\varepsilon}_j)$ pairs.

These results are compared with uniform sampling, which, given a specified size $n$, uniformly sample $n$ rows of $A$ to form a coreset $B$. For each $n$, we run 100 independent trials, calculating the empirical distortion parameter $\tilde{\varepsilon} = \max_{x \in X} |\|Bx\|_2^2 / \|Ax\|_2^2 - 1|$ as before. The mean and standard deviation of $\tilde{\varepsilon}$ for each $n$ are then plotted.

The results are shown in Figures 1. For the energy dataset, our coresets consistently achieve a much smaller distortion than the preset value of $\varepsilon = 0.25$. Additionally, the distortion is significantly lower than that of uniform sampling, with the mean approximately 40%–50% smaller and also about one standard deviation smaller. About 10% of the rows suffice to yield a subspace embedding with a distortion parameter of at most 0.15. For the Emission dataset, we see again that the distortion is much lower than the preset value of $\varepsilon = 0.25$. The mean distortion is about one standard deviation smaller than that of uniform sampling. About 5% of the rows achieve a distortion parameter of at most 0.1.

**Solving Optimization Problem** Now we apply our coresets to robust regression. Recall that, given $A \in \mathbb{R}^{n \times d}$ and $b \in \mathbb{R}^n$, robust regression seeks to solve $\min_{x \in \mathbb{R}^d} F^{(m)}(Ax - b)$, where $F(u)$ is the sum of $u_i^2$, excluding the $m$ largest coordinates (in absolute value).

To the best of our knowledge, the only algorithms for ro-

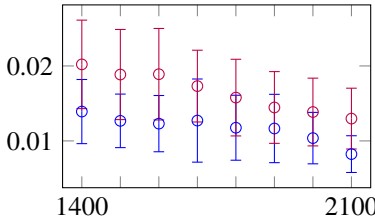 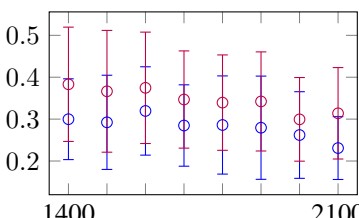 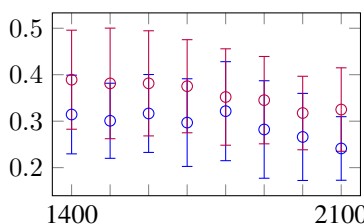

Figure 2: Robust regression on the Energy dataset with $m = 10$. Subplots show the relative error of the following quantities. Left: robust loss function $F^{(m)}$, Middle: $\ell_\infty$-norm, Right: $\ell_2$-norm.

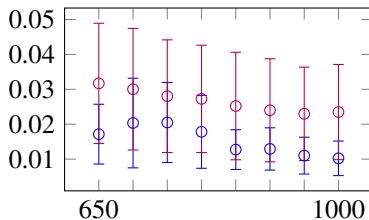 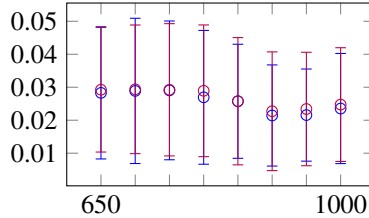 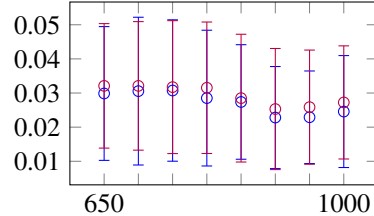

Figure 3: Robust regression on the Emission dataset with $m = 10$. Subplots show the relative error of the following quantities. Left: robust loss function $F^{(m)}$, Middle: $\ell_\infty$-norm, Right: $\ell_2$-norm.

bust $\ell_2$-regression with theoretical runtime guarantees that are substantially better than $\binom{n}{m}$ are that of Simonov et al. (2019) and our Algorithm 4. However, both remain computationally expensive, with runtimes exponential in $d^2$ and $m/\varepsilon$, respectively. The robust $\ell_2$-regression problem, also known as the least trimmed squares, has a long history of research. A popular heuristic approach is FastLTS[3] (Rousseeuw & van Driessen, 2006), which, for example, is implemented in Maple's statistics package (MapleSoft). We use FastLTS in our experiments.

We perform 1000 trials with different coresets, solving the regression problem using FastLTS for each coreset, yielding solutions $\tilde{x}_j$ for $j = 1, \ldots, 1000$. We also solve the original regression problem on the full data matrix using FastLTS to obtain a solution $\hat{x}$. Next, we evaluate the approximation of the objective function $|F^{(m)}(A\tilde{x}_j - b)/F^{(m)}(A\hat{x} - b) - 1|$ as well as the approximation of the solution in $\ell_2$ and $\ell_\infty$ norms: $\|\tilde{x}_j - \hat{x}\|_2/\|\hat{x}\|_2$ and $\|\tilde{x}_j - \hat{x}\|_\infty/\|\hat{x}\|_\infty$. Finally, we plot these quantities against the size of the coresets (rounded to the nearest multiple of 100 for the Energy dataset and the nearest multiple of 50 for the Emission dataset) and compare with the results using uniform sampling. We also report the runtimes[4] on our coresets of different sizes (including computing the coresets) as well as for the entire dataset. The runtime for the entire dataset is averaged over 10 trials.

The results for the Energy dataset are presented in Figure 2

and Table 1. We observe that the runtimes on coresets are substantially smaller. Additionally, our coresets significantly outperform uniform sampling in terms of the objective function, achieving lower mean relative errors and smaller standard deviations. Our coresets achieve a mean relative error of the objective function value in the range of 0.01 and 0.02 with coreset sizes of only 7.1% to 10.6% of the full dataset, significantly outperforming uniform sampling by at least one standard deviation. While the objective function is well-approximated, the solution approximation shows a larger relative error of approximately 0.25–0.3. We hypothesize that this is partly due to the large condition number of the data matrix ($\approx 2878$).

The results for robust regression on the Emission dataset are presented in Figure 3 and Table 1. As with the Energy dataset, we observe a substantial reduction in runtimes when using coresets. Once again, our coresets significantly outperform uniform sampling in terms of the objective function. The mean relative error of the objective function decreases from 0.02 to 0.01 as the coreset size increases from 2% to 2.7% of the data size. The solution error in both $\ell_\infty$ and $\ell_2$ norms are similar for both our coresets and uniform sampling, with mean relative errors around 0.03. We hypothesize that this is partly due to $\hat{x}$ having predominantly small coordinates except for one, making it relatively easy to approximate. However, small differences in the approximation can result in much larger differences in the objective function value.

---

[3]FastLTS is theoretically guaranteed to converge to the optimal solution but the convergence rate not understood. In practice, the parameters are chosen heuristically.

[4]All experiments were run on a machine with an Intel i5-1165G7 @ 2.80GHz CPU and 16 GB memory using Python version 3.12.8.

## Acknowledgements

The authors would like to thank the anonymous reviewers for their helpful suggestions, particularly regarding the expanded discussion on related work. C.W. In is supported by an NTU research scholarship. Y. Li is supported in part by, and X. Xuan is supported by, Singapore Ministry of Education AcRF Tier 2 grant MOE-T2EP20122-0001. D.P. Woodruff is supported in part by a Simons Investigator Award and Office of Naval Research (ONR) award number N000142112647.

## Impact Statement

This paper presents work whose goal is to advance the field of Machine Learning. There are many potential societal consequences of our work, none which we feel must be specifically highlighted here.

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

# A. Proof of Lemma 5.2

Without loss of generality, we may assume that $A$ has orthonormal columns. By the stardard proof for approximate regression such as that of Theorem 2.16 in (Woodruff, 2014), it suffices to assume that $A$ has orthonormal columns and show that with probability at least 0.98, (i) $\|A^\top S^\top SA - I\|_{op} \leq 1/2$ and (ii) $\|A^\top S^\top Sv\|_2 \leq \sqrt{\varepsilon}\|v\|_2$ for a specific vector $v \in \mathbb{R}^n$ such that $A^\top v = 0$.

We first show (i). Let $a_i$ denote the $i$-th row of $A$ (viewed as a column vector), then

$$A^\top S^\top SH - I = A^\top S^\top SH - A^\top A = \sum_{i=1}^{n} \xi_i a_i a_i^\top,$$

where $\xi_i$'s are i.i.d. Rademacher variables. We also have $\|a_i a_i^\top\|_{op} = \|a_i\|_2^2 \leq 1/r$ and $\|\sum_i (a_i a_i^\top)^2\|_{op} = \|\sum_i a_i(a_i^\top a_i)a_i^\top\|_{op} \leq (1/r)\|\sum_i a_i a_i^\top\|_{op} = (1/r)\|I\|_{op} = 1/r$. It follows from matrix Bernstein inequality that

$$\Pr\left\{\|A^\top S^\top SH - I\|_{op} > \frac{1}{2}\right\} \leq 2de^{-cr} \leq 0.01$$

(where $c > 0$ is an absolute constant), provided that $r \gtrsim \log d$.

Next we show (ii). Let $\{\eta_i\}$ be i.i.d. uniform variables on $\{0, 1\}$. We can then write

$$\begin{aligned}
\|A^\top S^\top Sv\|_2^2 &= \sum_{i=1}^{d}\left(\sum_j 2\eta_j a_{ji}v_j\right)^2 \\
&= 4\sum_i\sum_{j,\ell}\eta_j\eta_\ell a_{ji}a_{\ell i}v_j v_\ell \\
&= 4\sum_{j,\ell}\eta_j\eta_\ell\langle a_j, a_\ell\rangle v_j v_\ell \\
&= 4\sum_{j,\ell}\frac{\xi_j+1}{2}\cdot\frac{\xi_\ell+1}{2}\cdot\langle a_j, a_\ell\rangle v_j v_\ell \\
&= \sum_{j,\ell}\xi_j\xi_\ell\langle a_j, a_\ell\rangle v_j v_\ell + \sum_j \xi_j\left\langle a_j v_j, \sum_\ell a_\ell v_\ell\right\rangle \\
&\quad + \sum_\ell \xi_\ell\left\langle\sum_j a_j v_j, a_\ell v_\ell\right\rangle + \left\langle\sum_j a_j v_j, \sum_\ell a_\ell v_\ell\right\rangle.
\end{aligned}$$

Since $A^T v = 0$, we know that $\sum_j a_j v_j = 0$ and so

$$\|A^\top S^\top Sv\|_2^2 = \sum_{j,\ell}\xi_j\xi_\ell\langle a_j, a_\ell\rangle v_j v_\ell,$$

which is a quadratic form w.r.t. Rademacher variables $\xi_i$ and

$$\mathbb{E}\|A^\top S^\top Sv\|_2^2 = \sum_{j,\ell}\mathbb{E}(\xi_j\xi_\ell)\langle a_j, a_\ell\rangle v_j v_\ell = \sum_j\langle a_j, a_j\rangle v_j^2 \leq \frac{\|v\|_2^2}{r}$$

Let $G$ be a matrix with $G_{j,\ell} = \langle a_j, a_\ell\rangle v_j v_\ell$. We can calculate

$$\|G\|_F^2 = \sum_{j,\ell}\langle a_j, a_\ell\rangle^2 v_j^2 v_\ell^2 \leq \sum_{j,\ell}\|a_j\|^2\|a_\ell\|^2 v_j^2 v_\ell^2 \leq \frac{1}{r^2}\left(\sum_j v_j^2\right)^2 = \frac{1}{r^2}\|v\|_2^4$$

and

$$\|G\|_{op} = \|A^\top(\mathrm{diag}(v))^2 A\|_{op} = \sup_{\|x\|_2=1}\|\mathrm{diag}(v)Ax\|_2^2$$

$$\leq \|v\|_2^2 \sup_{\|x\|_2=1} \max_i \langle a_i, x \rangle^2$$

$$\leq \|v\|_2^2 \sup_{\|x\|_2=1} \max_i \|a_i\|_2^2 \|x\|_2^2$$

$$\leq \frac{1}{r} \|v\|_2^2$$

It follows from Hanson-Wright inequality that

$$\Pr\left\{\left|\|A^\top S^\top Sv\|_2^2 - \mathbb{E}\|A^\top S^\top Sv\|_2^2\right| \geq t\right\} \leq 2\exp\left(-c\min\left\{\frac{t^2}{\|v\|_2^4/r^2}, \frac{t}{\|v\|_2^2/r}\right\}\right)$$

Setting $t = C\|v\|_2^2/r$ (where $C$ is some absolute constant), we see that with probability at least 0.99,

$$\|A^\top S^\top Sv\|_2^2 \leq \frac{\|v\|_2^2}{r} + \frac{C\|v\|_2^2}{r} \leq \varepsilon\|v\|_2^2,$$

provided that $r \geq (C+1)/\varepsilon$, establishing (ii).

## B. Proof of Theorem 5.3

We present the algorithm in Algorithm 5.

---

**Algorithm 5** RobustPCA$(A, \varepsilon, m)$

---

**Input:** $A \in \mathbb{R}^{n \times d}$, parameters $\varepsilon$ and $m$
**Output:** $(1+\varepsilon)$-approx. solution to $\min_{U \in \mathcal{U}} F^{(m)}(AUU^\top - A)$
 1: $(D, \{w_i\}) \leftarrow \text{Coreset}(A, \varepsilon, m)$ {Each row $D_{i,*} \in \mathbb{R}^d$ is associated with weight $w_i$}
 2: Rescale each row of $D$ as $D_{i,*} \leftarrow \sqrt{w_i} D_{i,*}$ and set all weights to be 1
 3: $r \leftarrow O(\log d + 1/\varepsilon)$
 4: Duplicate each row of $D$ by $r$ times and rescale by $\frac{1}{\sqrt{r}}$, yielding $D'$
 5: $L \leftarrow 2^{O(rm)}$
 6: $X \leftarrow \emptyset$
 7: **for** $i = 1, 2, \ldots, L$ **do**
 8: $\quad$ $S \leftarrow$ random diagonal matrix with i.i.d. diagonal elements such that $S_{ii}$ is uniform on $\{0, \sqrt{2}\}$
 9: $\quad$ Compute $U' = \arg\min_{U \in \mathcal{U}} \|SD'UU^\top - SD'\|_F$
10: $\quad$ $X \leftarrow X \cup \{U'\}$
11: **end for**
12: Compute $\tilde{U} \leftarrow \arg\min_{U \in X} F^{(rm)}(D'UU^\top - D')$
13: **Return** $\tilde{U}$

---

To begin, we can write $\|AX - A\|_F^2$ as

$$\|AX - A\|_F^2 = \sum_{i=1}^n \|a_i X - a_i\|_2^2 =: \sum_{i=1}^n f_i(X)$$

where $a_i$ denotes the $i$-th row of $A$. We first show that the sensitivities for $f_i(X)$ is the $i$-th leverage score of $A$. Let $A = QR$ be the QR decomposition, where $Q \in \mathbb{R}^{n \times d}$ has orthonormal columns and $R$ is invertible, then

$$\sigma_{f_i} = \sup_{X:AX \neq 0} \frac{\|e_i^\top QRX\|_2^2}{\|QRX\|_F^2} = \sup_{X:QX \neq 0} \frac{\|e_i^\top QX\|_2^2}{\|QX\|_F^2}.$$

Since $Q$ has orthonormal columns, we know that $\|QX\|_F^2 = \|X\|_F^2$. Also, $\|e_i^\top QX\|_2^2 \leq \|e_i^\top Q\|_2^2 \|X\|_2^2$, thus

$$\sigma_{f_i} \leq \|e_i^\top Q\|_2^2,$$

where the equality is attained when $X = \begin{pmatrix} Q^\top e_i & Q^\top e_i & \cdots & Q^\top e_i \end{pmatrix}$. Therefore, $\sigma_{f_i} = \|e_i^\top Q\|_2^2$, which is exactly the $i$-th leverage score of $A$. This justifies Line 1 of the algorithm.

We also need to show an analogy of Lemma 5.2 for approximate PCA, which we state formally below.

**Lemma B.1.** *Suppose that $A \in \mathbb{R}^{n \times d}$ and its leverage scores are bounded by $1/r$. Let $S \in \mathbb{R}^{n \times n}$ be a random diagonal matrix, where $S_{ii} = \sqrt{2}$ with probability $1/2$ and $S_{ii} = 0$ with probability $1/2$. Let $U^* = \arg\min_{U \in \mathcal{U}} \|AUU^\top - A\|_F$ and $\tilde{U} = \arg\min_{U \in \mathcal{U}} \|SAUU^\top - SU\|_2$.*

*When $r \gtrsim d \log d + 1/\varepsilon$, it holds with probability at least $0.98$ that*

$$\|A\tilde{U}\tilde{U}^\top - A\|_F \leq (1+\varepsilon)\|AU^*(U^*)^\top - A\|_F.$$

*Proof.* The proof is nearly identical to that of Lemma 5.2 except that we need now to verify that $\|A^\top S^\top SV\|_F \leq \sqrt{\varepsilon}\|V\|_F$ for a specific matrix $V \in \mathbb{R}^{n \times d}$ such that $A^\top V = 0$. Write $V = \begin{pmatrix} v_1 & v_2 & \cdots & v_d \end{pmatrix}$, we have $\|A^\top S^\top SV\|_F^2 = \sum_{i=1}^d \|A^\top S^\top S v_i\|_2^2$. It follows from the calculations in the proof of Lemma 5.2 that

$$\|A^\top S^\top SV\|_F^2 = \sum_{j,\ell} \xi_j \xi_\ell \langle a_j, a_\ell \rangle \sum_i (v_i)_j (v_i)_\ell.$$

Thus,

$$\mathbb{E}\|A^\top S^\top SV\|_F^2 \leq \sum_i \frac{\|v_i\|_2^2}{r} = \frac{\|V\|_F^2}{r}.$$

Now we can write $G = \sum_i G_i$, where $(G_i)_{j,\ell} = \langle a_j, a_\ell \rangle (v_i)_j (v_i)_\ell$ and so

$$\|G\|_F^2 \leq \frac{1}{r^2} \sum_i \left( \sum_j (v_i)_j^2 \right)^2 \leq \frac{1}{r^2} \left( \sum_i \sum_j (v_i)_j^2 \right)^2 = \frac{1}{r^2} \|V\|_F^4$$

$$\|G\|_{op} \leq \sum_i \|G_i\|_{op} \leq \frac{1}{r} \sum_i \|v_i\|_2^2 = \frac{1}{r} \|V\|_F^2.$$

The desired result follows from Hanson-Wright inequality as in the proof of Lemma 5.2. $\square$

The remainder of the proof is nearly identical to that of Theorem 5.1, so we just give a sketch below. Let

$$U^* = \min_{U \in \mathcal{U}} F^{(m)}(AUU^\top - A), \quad U^\sharp = \min_{U \in \mathcal{U}} F^{(m)}(DUU^\top - D).$$

Define $J$ to be the set of indices of the largest $m$ rows (in $\ell_2$ norm) of $DUU^\top - D$ and $J'$ to be the $rm$ indices in $D'UU^\top - D'$ that correspond to the indices in $J$.

Consider the event $\mathcal{E}$ that $SD'$ contains none of the rows of indices in $J'$. With probability at least $0.99$, this event happens in at least one of the $L$ trials.

Consider the trial in which $\mathcal{E}$ happens. Let $D''$ be the resulting matrix from $D'$ after removing rows of indices in $J'$. By the preceding lemma and our choice of $r$, it holds with probability at least $0.98$ that

$$\|D''U'(U')^\top - D''\|_F^2 \leq (1+\varepsilon)^2 F^{(rm)}(D'U^\sharp(U^\sharp)^\top - D').$$

It follows that

$$F^{(m)}(D\tilde{U}\tilde{U}^\top - D) = F^{(rm)}(D'\tilde{U}\tilde{U}^\top - D') \leq (1+\varepsilon)^2 F^{(rm)}(D'U^\sharp(U^\sharp)^\top - D') = (1+\varepsilon)^2 F^{(m)}(DU^\sharp(U^\sharp)^\top - D).$$

Using the coreset properties, we have

$$F^{(m)}(A\tilde{U}\tilde{U}^\top - A) \leq \frac{1}{1-\varepsilon} F^{(m)}(D\tilde{U}\tilde{U}^\top - D) \leq \frac{(1+\varepsilon)^2}{1-\varepsilon} F^{(m)}(DU^\sharp(U^\sharp)^\top - D)$$

$$\leq \frac{(1+\varepsilon)^2}{1-\varepsilon} F^{(m)}(DU^*(U^*)^\top - D)$$

$$\leq \frac{(1+\varepsilon)^3}{1-\varepsilon} F^{(m)}(AU^*(U^*)^\top - A)$$

$$\leq (1+7\varepsilon) F^{(m)}(AU^*(U^*)^\top - A)$$

This completes the proof of correctness.

The analysis of the failure probability and the runtime is identical to the proof of Theorem 5.1, where we compute the SVD of $SD'$ to find $U'$.

## C. Partial Removal Model

In our definition, the robust loss function removes $m$ functions regardless of their weights, which we will refer to as the full removal model. In clustering (Huang et al., 2023b; 2025), an alternative definition has been used, which allows partial removal. Instead of removing exactly $m$ functions, this definition removes a total weight of $m$. Formally, let $\mathcal{F} = \{(f_i, \omega_i) \mid i \in [n]\}$ be a weighted set of functions. The robust loss function is then defined as

$$\mathcal{L}^{(m)}(\mathcal{F}; x) = \min_{\substack{w': \\ 0 \leq w' \leq w, \\ \|w - w'\|_1 \leq m}} \sum_{i=1}^{n} w_i' f_i(x).$$

Here, we write $w' \leq w$ for two vectors $w, w' \in \mathbb{R}^n$ to mean $w_i' \leq w_i$ for every $i \in [n]$.

Our main result is the following theorem, which shows that our coreset algorithm also works under this partial removal model.

**Theorem C.1.** *For every $\varepsilon \in (0, \frac{1}{4})$ and $m \in [n]$, the output of Algorithm 3 is also an $(\varepsilon, m)$-coreset for $F$ under partial removal model with constant probability.*

*Proof.* As in the proof of Theorem 4.1, we condition on the event that all contributing functions are in $S$, which happens with constant probability.

Let $P$ denote the output of $\mathrm{Coreset}(F, \varepsilon, m)$. We shall show that $P$ is an $(\varepsilon, m)$-coreset under the partial removal model; that is, for every $x \in \mathbb{R}^d$ and $t \in \{0, \cdots, m\}$, it holds that $\mathcal{L}^{(t)}(P; x) \in (1 \pm C\varepsilon) \cdot \mathcal{L}^{(t)}(A; x)$ for some absolute constant $C > 0$.

We first prove that $\mathcal{L}^{(t)}(P; x) \leq (1 + C\varepsilon) \cdot \mathcal{L}^{(t)}(A; x)$. Let $L_x$ denote the set of outliers[5] excluded by $\mathcal{L}^{(t)}(A; x)$, then $|L_x| \leq t$. Let $\tilde{L}_x = \{(f, 1) \mid f \in L_x\}$. We claim that it suffices to prove that

$$\sum_{(f,1) \in \tilde{S} \setminus \tilde{L}_x} f(x) + \sum_{(f, \omega_f) \in P \setminus \tilde{S}} \omega_f \cdot f(x) \leq (1 + 3\varepsilon) \, \mathcal{L}^{(t)}(A; x). \tag{2}$$

To see this, notice that the total weight of $\tilde{L}_x$ is at most $1 \cdot |L_x| \leq t$, thus the left-hand-side of (2) is at least $\mathcal{L}^{(t)}(P; x)$.

To prove (2), we begin by noting that

$$\sum_{(f,1) \in \tilde{S} \setminus \tilde{L}_x} f(x) = \sum_{f \in S \setminus L_x} f(x). \tag{3}$$

Next, it remains to show that

$$\sum_{(f, \omega_f) \in P \setminus \tilde{S}} \omega_f \cdot f(x) \leq (1 + \varepsilon) \sum_{f \in (A \setminus S) \setminus L_x} f(x) + 2\varepsilon \, \mathcal{L}^{(t)}(A; x), \tag{4}$$

which, when combined with (3), will give exactly (2).

To prove (4), we note that $P \setminus \tilde{S}$ is an $\varepsilon$-coreset of $A \setminus S$ by construction, so

$$\sum_{(f, \omega_f) \in P \setminus \tilde{S}} \omega_f \cdot f(x)$$

---

[5]Since all functions in $A$ have unit weights, we can assume without loss of generality that no outlier is partially removed.

$$\leq (1+\varepsilon)\sum_{f\in A\setminus S} f(x) \tag{5}$$

$$= (1+\varepsilon)\sum_{f\in(A\setminus S)\setminus L_x} f(x) + (1+\varepsilon)\sum_{f\in L_x\cap(A\setminus S)} f(x) \tag{6}$$

$$\leq (1+\varepsilon)\sum_{f\in(A\setminus S)\setminus L_x} f(x) + (1+\varepsilon)\cdot t\cdot \frac{\varepsilon}{m}\,\mathcal{L}^{(t)}(A;x) \tag{7}$$

$$\leq (1+\varepsilon)\sum_{f\in(A\setminus S)\setminus L_x} f(x) + 2\varepsilon\,\mathcal{L}^{(t)}(A;x). \tag{8}$$

Here, (5) follows from the coreset property, (6) from the relationship of sets, (7) from the definition of interesting functions and the conditioning on the event that all contributing functions have been added into $S$, and (8) from the fact that $\varepsilon\leq\frac{1}{4}<1$.

It remains to prove that $\mathcal{L}^{(t)}(A;x)\leq(1+C\varepsilon)\cdot\mathcal{L}^{(t)}(P;x)$ for some absolute constant $C>0$. Let $\omega'\in\mathbb{R}^{|P|}$ denote the weight vector induced by $\mathcal{L}^{(t)}(P;x)$, namely $\mathcal{L}^{(t)}(P;x)=\sum_{(f,\omega'_f)\in P}\omega'_f\cdot f$, $0\leq\omega'\leq\omega$, and $\|\omega-\omega'\|_1\leq t$.

Let $\omega''_f=\omega'_f$ for each $f\in S$ and $\omega''_f=1$ for each $f\in A\setminus S$. We note that $0\leq\omega''\leq1$ and $\|1-\omega''\|_1=\sum_{f\in S}(1-\omega'_f)\leq\|\omega-\omega'\|_1\leq t$. This means that $\omega''$ is a valid weight vector for $\mathcal{L}^{(t)}(A;x)$ and we can proceed as

$$\mathcal{L}^{(t)}(A;x)\leq\sum_{f\in A}\omega''_f\cdot f(x)$$

$$=\sum_{f\in S}\omega''_f\cdot f(x)+\sum_{f\in A\setminus S}\omega''_f\cdot f(x)$$

$$\leq\sum_{(f,1)\in\tilde{S}}\omega'_f\cdot f(x)+\sum_{f\in A\setminus S} f(x)$$

$$\leq\sum_{(f,1)\in\tilde{S}}\omega'_f\cdot f(x)+(1+2\varepsilon)\sum_{(f,\omega_f)\in P\setminus\tilde{S}}\omega_f\cdot f(x).$$

We claim that

$$\sum_{(f,\omega_f)\in P\setminus\tilde{S}}\omega_f\cdot f(x)\leq\sum_{(f,\omega_f)\in P\setminus\tilde{S}}\omega'_f\cdot f(x)+\varepsilon\,\mathcal{L}^{(t)}(A;x). \tag{9}$$

It then follows that

$$\mathcal{L}^{(t)}(A;x)\leq(1+2\varepsilon)\sum_{(f,\omega_f)\in P}\omega'_f\cdot f(x)+\varepsilon(1+2\varepsilon)\,\mathcal{L}^{(t)}(A;x),$$

which implies that (recalling that $\varepsilon\in(0,1/4)$)

$$\mathcal{L}^{(t)}(A;x)\leq\frac{1+2\varepsilon}{1-\varepsilon(1+2\varepsilon)}\,\mathcal{L}^{(t)}(P;x)$$

$$\leq(1+6\varepsilon)\,\mathcal{L}^{(t)}(P;x).$$

Now we prove our claim (9). We have that

$$\sum_{(f,\omega_f)\in P\setminus\tilde{S}}\omega_f\cdot f(x)$$

$$\leq\sum_{(f,\omega_f)\in P\setminus\tilde{S}}\omega'_f\cdot f(x)+\|\omega-\omega'\|_1\cdot\sup_{(f,\omega_f)\in P\setminus\tilde{S}} f(x)$$

$$\leq\sum_{(f,\omega_f)\in P\setminus\tilde{S}}\omega'_f\cdot f(x)+t\cdot\frac{\varepsilon}{m}\cdot\mathcal{L}^{(t)}(A;x)$$

$$\leq\sum_{(f,\omega_f)\in P\setminus\tilde{S}}\omega'_f\cdot f(x)+\varepsilon\cdot\mathcal{L}^{(t)}(A;x).$$

The proof is now complete. $\qquad\square$

As a corollary of the theorem, we obtain an improved coreset size for robust $k$-median under the partial removal model when the number of outliers is small. The current state-of-the-art $(\varepsilon, m)$-coreset under this model has size $m + \tilde{O}(k^2\varepsilon^{-4})$ (Huang et al., 2025). In the preceding section, we demonstrated that for robust $k$-median a coreset size under the full removal model, the coreset size is $O\left(\frac{mk}{\varepsilon} \log \frac{mk}{\varepsilon} + \frac{k}{\varepsilon^2} \cdot \min(k^{1/3}, \varepsilon^{-1})\right)$. By Theorem C.1, this result also holds for the partial removal model. This improves upon the result in (Huang et al., 2025) when $m = o(k\varepsilon^{-3})$.

## D. Lower Bound for Subspace Embedding

We prove a lower bound on the size of the robust coreset for subspace embedding. Let $t = \lfloor \frac{m}{\varepsilon} \rfloor$. We define $a_{(k-1)d+i} = e_i$ for every $k \in [t]$ and $i \in [d]$. Suppose that the matrix $A$ has rows $a_1, a_2, \ldots, a_{td}$ and we let $F_i(x) = |\langle a_i, x \rangle|^2$. Our main result is the following theorem.

**Theorem D.1.** *Any $(\varepsilon, m)$-coreset for $F(x) = \sum_{i=1}^{td} F_i(x)$ has size at least $\frac{md}{4\varepsilon}$.*

*Proof.* Let $D$ be an $(\varepsilon, m)$-coreset for $F$. Define $D_i = \{(\omega_f, f) \in D \mid f(x) = |\langle e_i, x \rangle|^2\}, i \in [d]$. So $D = \bigcup_{i=1}^d D_i$. Let $\tilde{F}(x) = \sum_{(\omega_f, f) \in D} \omega_f \cdot f(x)$.

Since $F(e_i) = t$ and $\sum_{(\omega_f, f) \in D} f(e_i) = \sum_{(\omega_f, f) \in D_i} \omega_f$, by the coreset property $\tilde{F}(x) \le (1 + \varepsilon) \cdot F(x)$, we have that,

$$\sum_{(\omega_f, f) \in D_i} \omega_f \le (1 + \varepsilon) \cdot t.$$

If $|D_i| \le \frac{m}{4\varepsilon}$, we know that

$$
\begin{aligned}
\tilde{F}^{(m)}(e_i) &\le \frac{|D| - m}{|D|} \cdot \sum_{(\omega_f, f) \in D_i} \omega_f \\
&\le \frac{\frac{m}{4\varepsilon} - m}{\frac{m}{4\varepsilon}} \cdot (1 + \varepsilon) \cdot t \\
&\le (1 - 3\varepsilon) \cdot t \\
&= (1 - 3\varepsilon) \cdot \frac{t}{t - m} \cdot (t - m) \\
&\le (1 - 3\varepsilon) \cdot \frac{\frac{m}{\varepsilon}}{\frac{m}{\varepsilon} - m} \cdot (t - m) \\
&= \frac{1 - 3\varepsilon}{1 - \varepsilon} \cdot F^{(m)}(e_i) \\
&\le (1 - 2\varepsilon) \cdot F^{(m)}(e_i).
\end{aligned}
$$

But the coreset property asserts that $\tilde{F}^{(m)}(e_i) \ge (1 - \varepsilon) \cdot F^{(m)}(e_i)$, which is a contradiction. So $|D_i| > \frac{m}{4\varepsilon}$ and $|D| = \sum_{i=1}^d |D_i| \ge \frac{md}{4\varepsilon}$. $\qquad \square$

