# OpenReview forum: "Robust Sparsification via Sensitivity"
_ICML.cc/2025/Conference — ICML 2025 poster_

### Official Review · Reviewer_YNPP · 2025-03-11

**Overall Recommendation:** 3

**Summary:**

The paper proposes a general framework for constructing ε-coresets for robust optimization problems of the form $\min_{x \in \mathbb{R}^d} F(x) = \sum_{i=1}^n F_i(x)$, where the robust version $F^{(m)}(x)$ aggregates all but the $m$ largest values of $F_i(x)$. This formulation is motivated by the need to handle outliers in machine learning tasks such as regression, PCA, and clustering. The authors develop an algorithm that constructs an ε-coreset of size $O(mT/\varepsilon \cdot \log(mT/\varepsilon)) + S$, where $T$ is the total sensitivity and $S$ is the size of a vanilla ε-coreset, assuming the latter exists. The approach leverages sensitivity sampling and a novel sensitivity flattening technique, yielding scalable algorithms with near-tight bounds for problems like $\ell_2$-subspace embeddings.

**Claims And Evidence:**

Yes

**Essential References Not Discussed:**

Please see "Other Strengths And Weaknesses" part.

**Experimental Designs Or Analyses:**

It proposes a general procedure that may extend vanilla coreset to coreset with outliers. The experimental part is clear.

**Methods And Evaluation Criteria:**

Yes

**Other Comments Or Suggestions:**

N/A

**Other Strengths And Weaknesses:**

1. Not very new problem. For example, the outlier problem as proposed in **Question 1.1** has been commonly studied, which can be traced back to [RL87].
2. Lack of novelty and more closely related papers should be referred.  The paper [WGD21] also studies robust coresets for  continuous-and-bounded learning, which **already includes applications like regression, PCA, and clustering, as covered in submitted paper**.  They share some similar technical ideas.

[RL87] Peter J. Rousseeuw and Annick Leroy. Robust Regression and Outlier Detection. Wiley, 1987

[WGD21] "Robust and fully-dynamic coreset for continuous-and-bounded learning (with outliers) problems." *Advances in Neural Information Processing Systems* 34 (2021): 14319-14331.

**Questions For Authors:**

Please see "Other Strengths And Weaknesses" part.

**Relation To Broader Scientific Literature:**

It may be useful to analyze large scale scientific data in physics, biology, and chemistry.

**Theoretical Claims:**

This paper is well-written, the proofs are clear, and seems sound to my knowledge.

---

> ### Author Rebuttal · Authors · 2025-04-01
>
> We thank the reviewer for the questions. Below are our responses.
>
>     Not very new problem. For example, the outlier problem as proposed in Question 1.1 has been commonly studied, which can be
>     traced back to [RL87].
>
> We agree that robust estimation [RL87] is classical work in the field. However, our work addresses the distinct and more recent challenge of constructing coresets for robust objectives. This specific problem of robust coreset construction is not addressed in [RL87], and the theory significantly lags behind that of standard (non-robust) coresets. Our paper makes novel contributions specifically to this direction.
>
>     Lack of novelty and more closely related papers should be referred. The paper [WGD21] also studies robust coresets for
>     continuous-and-bounded learning, which already includes applications like regression, PCA, and clustering, as covered in
>     submitted paper. They share some similar technical ideas.
>
> We thank the reviewer for pointing out [WGD21]. We will add a detailed discussion comparing our work in the revised manuscript. There are crucial differences:
>
> (1) The definition of a coreset in [WGD21] differs fundamentally from ours. Their coreset preserves the loss function within a ball, closer to the notion of a "weak coreset" in the literature. In contrast, a "strong coreset" preserves the loss function for all parameters, which is the standard, more powerful notion of a coreset in related work and is significantly harder to achieve in the robust setting. It also allows for arbitrary additional constraints in the optimization problem as it preserves the loss function for all parameters. Our work is the first to achieve this for a general class of loss functions.
>
> (2) The approximation guarantee in [WGD21] requires relaxing $m$ (allowing more points to be removed than specified) or depends on an implicit problem-dependent parameter $\epsilon_0$. Our guarantee holds for the precise given $m$ without such a relaxation or hidden dependencies.
>
> (3) While both [WGD21] and our algorithm use two stages, the techniques differ substantially due to the different definitions of coresets. The first stage of [WGD21] identifies all local potential outliers (with respect to a fixed solution), while our first stage identifies all global "contributing functions" (regardless of the solution) through a novel process that combines uniform sampling with sensitivities.
> While both methods use a vanilla coreset in the second stage, we introduce an additional sensitivity-flattening step to satisfy the requirements of a strong coreset, which is absent in [WGD21].
>
> Therefore, while related in appearance, [WGD21] tackles a different (weaker) coreset definition with distinct techniques. Our work provides the first strong robust coreset for this general setting, marking a significant theoretical advance.

---

> > ### Comment · Reviewer_YNPP · 2025-04-05
> >
> > I thank the authors for the detailed response to my concerns. The rebuttal addressed part of my concerns. Also, given the positive grades from other reviewers, I would like to raise my score to weak accept.

---

### Official Review · Reviewer_L5ZF · 2025-03-12

**Overall Recommendation:** 4

**Summary:**

This work studies the coreset for robust optimization problems, where the loss function is defined to allow the removal of the highest $m$ costs. Research on robust coreset is relatively limited compared to the vanilla version. For functions with total sensitivity $T$, using the vanilla coreset algorithm and sensitivity oracle, the authors introduce a meta-algorithm for constructing a robust coreset of additional size $O(Tm/\varepsilon\log (Tm/\varepsilon))$.

Roughly speaking, the robust coreset consists of two parts. The first part is a set of contributing functions with unit weight, achieved through  $O(m\log(Tm/\varepsilon))$ rounds of sampling and selection. The second part is a refined coreset of the remaining functions. The refinement is necessary because the weight of a function in the vanilla coreset may be too large, potentially violating the definition of a robust coreset.

In addition to the meta-algorithm, the robust coreset offers a pathway to develop approximation algorithms for robust optimization problems. This paper presents improved algorithms for (robust) regression and PCA by demonstrating bounded total sensitivity.

**Claims And Evidence:**

Theoretical paper. All claims are supported by corresponding proofs.

**Essential References Not Discussed:**

[1] also explores the robust coreset within a broader context. And their robust coreset also consists of a vanilla coreset and a  sampling-based component of size $O(m/\varepsilon)$ (can be improved with bounded doubling dimension). However, in their study, the coreset property only holds for local $x$.  And it includes a relaxiation of $m$, which aligns with the result in [2]. Regrading the generality of this paper, I am curious whether their work can be incorporated into your framework as a special case.


---
[1] Robust and Fully-Dynamic Coreset for Continuous-and-Bounded Learning (With Outliers) Problems.
[2] Near-optimal Coresets for Robust Clustering.

**Experimental Designs Or Analyses:**

Yes. The work aligns with the consistent experimental style of the coreset research.

**Methods And Evaluation Criteria:**

In the context of a theoretical paper, the primary evaluation criterion is the correctness of the main theorem. As for the experimental component, the results are meaningful and make sense in the field of coreset.

**Other Comments Or Suggestions:**

- Line 098, “The total sensitivity of $\mathcal{F}$” should be “The sensitivity of $\mathcal{F}$”
- Line 159, “A function $f$ is” , missing “ $\in A$ “
- Line 217, $F$ in the Uniform() should be $A$

**Other Strengths And Weaknesses:**

## Strengths
- The algorithm is concise and easy to implement, making it accessible for further development. The proof is clear and appears to be correct, based on my assessment. Its clarity enhances the understanding of the algorithm's validity and effectiveness.
- The assumption of sensitivities is reasonable, as it is reasonable when regarding real-world downstream tasks such as PCA and regression.

## Weaknesses
From a practical standpoint, the robust coreset is less critical. This is because robust algorithms for optimization problems with outliers often lack guaranteed approximation, leading to a reliance on heuristic methods for solutions. In such cases, compressing the data as a preprocessing step becomes less significant, particularly when the compression method is relatively complex and time-consuming.

**Questions For Authors:**

The effectiveness of the proposed algorithm depends on a bound of total sensitivity $T$ and a running time of sensitivity oracle $t_1$. In other words, the algorithm is suitable for such problems that (1) total sensitivity is not large (2) The time of computing sensitive is limited. Is there a large class of problems satisfying such condition besides the cases presented?

**Relation To Broader Scientific Literature:**

not sure

**Theoretical Claims:**

Yes. i have checked all the proofs in the main body of the paper

---

> ### Author Rebuttal · Authors · 2025-04-01
>
> We thank the reviewer for the questions. Below are our responses.
>
>     [1] also explores the robust coreset within a broader context. And their robust coreset also consists of a vanilla
>     coreset and a sampling-based component of size $O(m/\varepsilon)$ (can be improved with bounded doubling dimension).
>     However, in their study, the coreset property only holds for local $x$. And it includes a relaxiation of $m$, which
>     aligns with the result in [2]. Regrading the generality of this paper, I am curious whether their work can be
>     incorporated into your framework as a special case.
>
>     [1] Robust and Fully-Dynamic Coreset for Continuous-and-Bounded Learning (With Outliers) Problems. [2] Near-optimal Coresets for Robust Clustering.
>
> We generally agree with the reviewer's comment on [1]. We note that the framework in [1] is based on assumptions that the loss functions satisfy the Lipschitz condition and the boundedness property, which are not assumed in our paper. Therefore, the two papers are not directly comparable. We also refer to our response to the second point of Reviewer YNPP for a detailed comparison.
>
>     From a practical standpoint, the robust coreset is less critical. This is because robust algorithms for optimization
>     problems with outliers often lack guaranteed approximation, leading to a reliance on heuristic methods for solutions.
>     In such cases, compressing the data as a preprocessing step becomes less significant, particularly when the compression
>     method is relatively complex and time-consuming.
>
> We understand the concern regarding practicality, especially when heuristics are common. However, our robust coreset offers a significant advantage: it enables the use of, or accelerates, algorithms on datasets previously too large. For example, as detailed in the appendix, applying FastLTS for trimmed least squares to the full dataset took 7.543 seconds. By using our coreset, sized at only approximately 2.7\% of the original data, the same FastLTS implementation achieved good accuracy in just 0.4 seconds.
>
> More generally, for computationally intensive algorithms with potentially prohibitive runtimes (e.g., scaling as $\binom{n}{m} = \exp(O(m\log\frac{n}{m}))$ in the worst case),
> we reduce the effective input size from $n$ to $O(md)$ (assuming $\epsilon$ is a constant), making these algorithms more feasible for larger-scale problems.
>
>     The effectiveness of the proposed algorithm depends on a bound of total sensitivity  T and a running time of sensitivity
>     oracle t1. In other words, the algorithm is suitable for such problems that (1) total sensitivity is not large (2) The
>     time of computing sensitive is limited. Is there a large class of problems satisfying such condition besides the cases
>     presented?
>
> While current prominent applications of sensitivity sampling focus on regression, PCA, and clustering, the framework itself is general.
> We emphasize that approximate sensitivities, rather than exact ones, are sufficient for coreset construction, making this approach adaptable to a wider range of problems where exact sensitivities might be intractable. Just as efficient algorithms were developed for approximating leverage scores, we anticipate similar progress for sensitivity estimation in other problems, and hope our work encourages such research by demonstrating the utility of sensitivity in the robust setting.
>
>     Line 098, “The total sensitivity of $\mathcal{F}$” should be “The sensitivity of $\mathcal{F}$”
>     Line 159, “A function $f$ is” , missing “$\in A$"
>     Line 217, $F$ in the Uniform() should be $A$
>
> We agree with the reviewer on all these points. We will correct them in the revised version.

---

> > ### Comment · Reviewer_L5ZF · 2025-04-02
> >
> > Thank you for the response. After reading your discussions with other reviewers, I agree with the distinct contributions and improvements this work presents compared to existing research. As for me, the primary significance lies in being the first to achieve "global" robust coreset without requiring relaxation in the number of outliers.  I think it is necessary to clarify this claim in the paper if it was verified.
> >
> > Also, as pointed out by other reviewers, the current version lacks a thorough review of robust coreset, which is only briefly mentioned in the introduction section. Actually, research on robust coreset began very early. In addition to [1, 2], references [3] and [4] also explore robust coreset from certain perspectives. A more detailed comparison should be included in the related work section.
> >
> >
> >
> > [3] SOTC’11 A unified framework for approximating and clustering data
> >
> > [4] FOCS’18 ε-Coresets for Clustering (with Outliers) in Doubling Metrics

---

> > > ### Author Response · Authors · 2025-04-04
> > >
> > > We thank the reviewer for pointing out additional related work. In the revised version, we will include a more detailed discussion comparing our results with existing literature. The two papers [3] and [4] mentioned by the reviewer provide bicriteria guarantees. In addition, the coreset in [5] has an exponential size, which is significantly worse than the standard vanilla coreset sizes.
> > >
> > > [5] Dan Feldman, Leonard J. Schulman. Data reduction for weighted and outlier-resistant clustering. SODA 2012.

---

### Official Review · Reviewer_tJCf · 2025-03-12

**Overall Recommendation:** 3

**Summary:**

This paper studies a robust version of the $\epsilon$-coreset construction for a function class $\mathcal{F}$. Specifically, if we assume that there are $m$ outliers in $\mathcal{F}$, the goal is to construct a coreset such that it will always be an $\epsilon$-coreset even if we remove up to $m$ largest functions, valued with respect to any model $x$. This definition is strong in the sense that, it needs to hold for any model $x$, as the input to the functions. The construction is built upon vanilla coreset construction which refers to the non-robust counterpart, which includes a uniform sampling stage and a refining stage. The uniform sampling tries to collect most of the functions with large sensitivity; while the refining stage verifies the rest, and include any function that escapes the first stage. The paper also shows the results can be used in many problems such as robust linear regression and robust PCA.

**Claims And Evidence:**

The claims are supported with clear and convincing evidence. The claims are clearly stated, the algorithms are clearly presented, and the proofs are well organized and written.

**Essential References Not Discussed:**

See Questions for Authors.

**Experimental Designs Or Analyses:**

I quickly scanned the experiments and found no issue.

**Methods And Evaluation Criteria:**

The method makes sense. As the paper studies the coreset problem by proposing a robust version to the vanilla $\epsilon$-coreset construction. The algorithm includes two stages: 1) uniform sampling to detect functions with large sensitivity; 2) constructing a vanilla coreset and refining it by adding more weight to functions with large sensitivity.

**Other Comments Or Suggestions:**

In the proof for Lemma 4.4, Line 212, given that the total sensitivity is bounded by $T$, why is it that there are at most $\frac{4T}{\epsilon}$ functions added into $D$ in each repetition? I didn't quite get this part. Why is this function unrelated to $m$?

**Other Strengths And Weaknesses:**

The paper is very solid and the proof is sound.

In Theorem 5.1, the running time of the algorithm for robust regression, it is $d^{O(m)}e^{O(m/\epsilon)} + O(nd)$, meaning that it is exponential in $m$. This is a large running time when $m$ is large, and may prohibit realistic application of the algorithm to practical problems.

**Questions For Authors:**

While this paper is compared to Simonov et al. 2019 for robust PCA, I wonder how do you compare it to more recent paper:

Nearly-Linear Time and Streaming Algorithms for Outlier-Robust PCA, by I. Diakonikolas, D. Kane, A. Pensia, T. Pittas (ICML 2023).

Specifically, the running time in this paper, and that of Simonov et al. 2019 is exponential in either $m$ or $d$, which is avoided in Diakonikolas et al. 2023. One reason might be that there seems no assumption of the $m$ outliers in this paper, while Diakonikolas et al. 2023 has distribution assumptions. Is this correct? Otherwise, the running time seems not reasonable. If possible, could you also comment on your current running time in terms of how optimal it is?

**Relation To Broader Scientific Literature:**

The result of this paper, can potentially be applied to many problems, such as regression, PCA, k-median problems, etc. As such, it can be used to build robust algorithms for many different machine learning problems.

On the other hand, this means the proposed technique is very general. Hence, if we were to solve a specific problem (for example, robust linear regression has a rich literature of its own), it might not be comparable to existing robust algorithms.

**Theoretical Claims:**

I checked most of the proofs in Section 4 (especially the key algorithm steps), and verified the application of these theorems in Section 5. All steps I checked are sound.

---

> ### Author Rebuttal · Authors · 2025-04-01
>
> We thank the reviewer for the questions. Below are our responses.
>
>     This is a large running time when m  is large, and may prohibit realistic application of the algorithm to practical problems.
>
> We acknowledge the exponential dependence on $m$. However, many existing robust algorithms, including practical ones like FastLTS or theoretical approaches, involve computations with worst-case complexities like $\binom{n}{m} = \exp(O(m\log\frac{n}{m}))$. Using our coresets, the runtime would become $\exp(O(m\log d))$ (assuming that $\epsilon$ is a constant). In typical large-scale scenarios where $n\gg d$ and $n\gg m$, this is a significant improvement in the base of the exponent.
>
>      In the proof for Lemma 4.4, Line 212, why is the number of functions added not dependent on m?
>
> The reviewer is correct to note the number of functions added in a single execution of Line 4 (Algorithm 3) does not explicitly depend on $m$. This line calls Algorithm 1, which adds functions whose sensitivity relative to the sampled set $B$ is at least $\epsilon/4$.
> By definition of total sensitivity, the sum of sensitivities is at most $T$, meaning at most $4T/\epsilon$ functions can satisfy this $\geq \epsilon/4$ condition.
> The threshold is $\epsilon/4$ to ensure that each contributing function (defined with $\epsilon/m$) is caught with sufficient probability ($1/(5m)$ as in Lemma 4.3), but the number of functions added in one execution of Algorithm 1 is bounded only in terms of $T$ and $\epsilon$.
> As explained above, the dependence on $m$ lies in the probability that each contributing function is captured, and this dependence enters the number of repetitions $R$ (Lines 2 and 3 of Algorithm 3).
>
>     Comparison with the paper "Nearly-Linear Time and Streaming Algorithms for Outlier-Robust PCA, by I. Diakonikolas, D. Kane, A. Pensia, T. Pittas (ICML 2023)."
>
> Yes, the reviewer is correct. The crucial difference is that Diakonikolas et al. (2023) achieve nearly-linear time by leveraging assumptions about the data distribution. Our work provides guarantees in the worst-case setting without such distributional assumptions, which inherently makes the problem harder and often necessitates complexities exponential in parameters like $m$ or $d$.
>
>     If possible, could you also comment on your current running time in terms of how optimal it is?
>
> Regarding optimality for robust PCA: Simonov et al. (2019) established that no polynomial-time algorithm exists under standard complexity assumptions, suggesting that exponential dependency on some parameters is likely necessary for worst-case guarantees. They provide a lower bound of $m^{\Omega(k)}$ and an upper bound of $n^{O(d^2)}$. Our coreset construction takes $d^{O(m)}$ time. While not directly comparable to their upper bound (better for small $m$, worse for large $m$), plugging our coreset (size $\approx md$ assuming a constant $\epsilon$) into their algorithm improves their runtime dependence on $n$, yielding roughly $(md)^{O(d^2)}$.
> This significantly reduces the base compared to $n^{O(d^2)}$ when $n\gg md$. While there remains a theoretical gap between our exponential runtime and known lower bounds, our approach offers a concrete improvement over existing worst-case algorithms by mitigating the dependence on $n$.

---

### Official Review · Reviewer_EHYA · 2025-03-14

**Overall Recommendation:** 4

**Summary:**

This manuscript shows that two simple conditions are sufficient for the existence of a small coreset for$F(m)$: $F(x)$ has a small vanilla coreset and has bounded total sensitivity. Then develops a general framework for constructing ε-coresets for several robust problems. Experiments on real-word datasets demonstrate that the coreset constructions are effective in approximating the loss function and considerably reduce the running time for robust regression problems while maintaining a good approximation of the objective function.

**Claims And Evidence:**

The claims and proofs in this manuscript are clear and explicit, and the writing style is organic unity, loose in form but focused in content, reaching the level of a professor.

**Essential References Not Discussed:**

The author is authoritative in the relevant field, so no other relevant literature is not cited.

**Experimental Designs Or Analyses:**

The experimental part of this article is relatively concise. I have two questions:

In the experimental part, this scheme is only compared with uniform sampling. I understand that core sets are carefully designed preferential sampling. So, are the results of such sampling better than those of random sampling?

The article assumes that the m to be removed is known (the form of the robust optimization problem defined in Section 1 uses m, and the input parameters of Algorithm 3 include m, m=10 given in your codes from). m is a parameter used to control the number of functions to be removed, which has not been effectively discussed.

**Methods And Evaluation Criteria:**

The method proposed in this manuscript is a new entry point for the construction of core sets and can well promote the development of related fields.

**Other Comments Or Suggestions:**

1. Practicality needs to be considered. If the greatest value of an algorithm improvement comes from the article itself, wouldn’t that be a pity?
2. Publish the code within an appropriate scope so that more people can comment, improve and use it.

**Other Strengths And Weaknesses:**

The paper is highly original, considering a wider range of robust core set selections and developing corresponding algorithms. It would be more conducive to promoting related research if the code was made public rather than limited to reviewers, but this is not necessary. The relevant arguments are clear and explicit. However, its importance and impact on the entire machine learning may be local and limited, especially when it comes to specific applications.

**Questions For Authors:**

The discussion about random sampling and m value in the experimental part are two important issues that affect my score. In addition, there are two more issues, but they are relatively unimportant compared to the above issues.
1、	With the robust formulation, has the context of facility been satisfactorily solved? What I mean is that we often take advantage of the loopholes in other people's theoretical research and continue to conduct theoretical research on the problem, but ignore the most real needs in the modeling process of this problem. Maybe this kind of theoretical research is not practical.
2、	Why was robust coresets first proposed under the condition of clustering? What are the essential differences between the PCA and regression promoted in this article and clustering?

**Relation To Broader Scientific Literature:**

As stated in the manuscript: In the robust case, coresets with similar size and performance to the vanilla case were not known until recently in the context of clustering with outliers. Beyond clustering, no other robust coresets have been proposed. So, this is an extension of previous research.

**Theoretical Claims:**

The theoretical derivation is logically rigorous, which is the author's greatest advantage and cannot be refuted. The proofs in the text and the appendix are rigorous and correct. As for practical applications, from the Impact Statement, it seems that this article does not care about.

---

> ### Author Rebuttal · Authors · 2025-04-01
>
> We thank the reviewer for the questions. Below are our responses.
>
>     this scheme is only compared with uniform sampling. I understand that core sets are carefully designed preferential
>     sampling. So, are the results of such sampling better than those of random sampling?
>
> We compared against uniform sampling as it is a standard and widely used baseline in the coreset literature. However, we acknowledge the reviewer's point regarding other sophisticated sampling methods. We will incorporate a comparison with leverage score sampling and a discussion in the revised manuscript.
>
>     The article assumes that the m to be removed is known. m is a parameter used to control the number of functions to be
>     removed, which has not been effectively discussed.
>
> Our formulation assumes a given $m$, consistent with the standard definition of problems like trimmed least squares and much of the existing robust coreset literature. Determining the value of $m$ is indeed a different and challenging problem, often tackled with heuristics that lack the strong guarantees our coreset framework enables. This falls outside the scope of this work, which is focused on guaranteed approximation for fixed $m$.
>
>     relative error of the experiments in the appendix is slightly worse than that in the main text
>
> We agree with this point. The observed difference in relative error is dataset-dependent. Specifically, the Emission dataset (appendix) is nearly 1.8 times the size of the Energy dataset (main text). Since we compared performance using coresets of similar sizes for both, it is understandable that the relative error would be slightly higher when applied to the much larger Emission dataset.
>
>     With the robust formulation, has the context of facility been satisfactorily solved? What I mean is that we often take
>     advantage of the loopholes in other people's theoretical research and continue to conduct theoretical research on the
>     problem, but ignore the most real needs in the modeling process of this problem. Maybe this kind of theoretical
>     research is not practical.
>
> While constructing the coreset is a preprocessing step, it directly enables the application of robust algorithms to much larger datasets. For instance, established algorithms for trimmed least squares like FastLTS, while considered practically effective, have worst-case runtimes depending exponentially on $m$ and polynomially on $n$. Our coreset significantly reduces this dependency on $n$, replacing it with a much smaller $md$ (our coreset size is $O(md)$, assuming $\epsilon$ is a constant). This allows applying such robust methods (with their existing guarantees or practical performance) in large-scale settings where they were previously intractable.
>
>     Why was robust coresets first proposed under the condition of clustering? What are the essential differences between the
>     PCA and regression promoted in this article and clustering?
>
> We note that the concept of a coreset was first introduced by Har-Peled and Mazumdar in their seminal work as a tool for addressing clustering problems. Over the past two decades, coresets for clustering have received considerable attention, making it unsurprising that robust coresets were initially developed in this context. Our work provides a unifying framework for constructing robust strong coresets (guaranteeing approximation for all $x$) applicable to regression, PCA, and potentially clustering. To our knowledge, prior work on robust coresets for regression and PCA either did not exist or considered different, weaker definitions (e.g., local guarantees, like [WGD21], as discussed further in response to Reviewer YNPP) which do not provide the same level of worst-case theoretical guarantees as our strong coreset definition.
>
> [WGD21] Zixiu Wang, Yiwen Guo, and Hu Ding. Robust and fully-dynamic coreset for continuous-and-bounded learning (with outliers) problems. Advances in Neural Information Processing Systems 34 (2021): 14319-14331.

---

> > ### Comment · Reviewer_EHYA · 2025-04-08
> >
> > Dear author, this is a field you are very familiar with, and your reply is self-consistent. I suggest you absorb the existing large language model to solve large-scale problems, then do some theoretical + engineering researches, applying theory to practice. But "hell is others", you can stick to doing your own research. Thank you.

---

### Decision · Program_Chairs · 2025-05-01

**Decision:**

Accept (poster)

**Comment:**

This paper shows that under bounded sensitivity, robustly construction of coresets can be reduced to noise-free case which has a rich literature. Such reduction is accompanied a natural 2-stage algorithm. Reviewers agree that the results are interesting and novel, and that the techniques are useful.